# Satellite observations reveal thirteen years of reservoir filling strategies, operating rules, and hydrological alterations in the Upper Mekong River Basin

Dung Trung Vu[1], Thanh Duc Dang[1,2], Stefano Galelli[1], and Faisal Hossain[3]

[1]Pillar of Engineering Systems and Design, Singapore University of Technology and Design, Singapore
[2]Department of Civil and Environmental Engineering, University of South Florida, Tampa, FL, USA
[3]Department of Civil and Environmental Engineering, University of Washington, Seattle, WA, USA
**Correspondence:** Stefano Galelli (stefano_galelli@sutd.edu.sg)

**Abstract.** The current situation in the Lancang–Mekong River Basin is emblematic of the issues faced by many transboundary basins around the world: riparian countries prioritize national water-energy policies and provide limited information on how major infrastructures are operated. In turn, such infrastructures and their management become a source of controversy. Here, we turn our attention to the Upper Mekong River, or Lancang, where a system of eleven mainstream dams controls about 55% of the annual flow to Northern Thailand and Laos. Yet, assessing their actual impact is a challenging task because of the chronic lack of data on reservoir storage and dam release decisions. To overcome this challenge, we focus on the ten largest reservoirs and leverage satellite observations to infer 13-year time series of monthly storage variations. Specifically, we use area-storage curves (derived from a Digital Elevation Model) and time series of water surface area, which we estimate from Landsat images through a novel algorithm that removes the effects of clouds and other disturbances. We also use satellite radar altimetry water level data (Jason and Sentinel-3) to validate the results obtained from satellite imagery. Our results describe the evolution of the hydropower system and highlight the pivotal role played by Xiaowan and Nuozhadu reservoirs, which make up to ∼85% of the total system's storage in the Lancang River Basin. We show that these two reservoirs were filled in about two years, and that their operations was marginally affected by the drought that occurred in the region in 2019-2020. Deciphering these operating strategies will help enrich existing monitoring tools and hydrological models, thereby supporting riparian countries in the design of more cooperative water-energy policies.

## 1 Introduction

In many transboundary river basins, conflicting dynamics between riparian countries are typically the result of different views on infrastructure development and management (Warner and Zawahri, 2012). Such dynamics are often compounded by the lack of transparency on how major infrastructure, such as dams, are operated. The situation in the Lancang–Mekong River

basin is no exception: during the past three decades, the basin has witnessed a rapid development of its hydropower fleet (Chowdhury et al., 2021), which has altered the hydrological regime (Dang et al., 2016; Räsänen et al., 2017) and changed the sediment budget (Kondolf et al., 2018; Binh et al., 2020), thereby degrading riverine ecosystems and threatening riparian communities (Sabo et al., 2017; Soukhaphon et al., 2021). In turn, these profound and ramified changes have challenged the relation between riparian countries (Wei et al., 2021). In this water-energy management 'mishmash', China plays a critical role. First, the river originates in the Tibetan Plateau and flows within the Chinese borders for about 2000 km, creating a natural power asymmetry with the other riparian countries (Kattelus et al., 2015). Second, China has built a limited number of dams—only eleven out of the hundred, or more, that currently punctuate the entire basin (Hecht et al., 2019). Yet, these few dams in the Upper Mekong River, or Lancang, have massive storage capacity ($\sim$42 km$^3$) and control a sizeable portion of the river discharge (about 55% of the average annual flow measured in Northern Thailand). Third, China has participated fairly weakly in transboundary water cooperation efforts, prioritizing bilateral cooperation to multi-country engagements, such as the Mekong River Commission (Kattelus et al., 2015; Williams, 2020). Fourth, China has yet to share detailed and comprehensive data on dam operations; agreements on data sharing and quality control are only at their infancy (Johnson, 2020). Because of these reasons, the Lancang's dams have become a source of controversy between China and downstream countries (IRN, 2002; Eyler and Weatherby, 2020; Kallio and Fallon, 2020). But to assess their actual impact and inform cooperative efforts, we must first quantify and understand how these dams have been operated.

There are at least two approaches available to tackle this challenge. The first one builds on the idea of generating data on reservoir inflow, storage, and release via simulation with a process-based hydrological-water management model; a solution recently explored for the Mekong Basin by Dang et al. (2020a), Yun et al. (2020), and Shin et al. (2020). Naturally, this is only a partial fix, since the simulation of water reservoir storage and operations still requires some basic information on design specifications and operational strategies. The second approach relies on satellite remote sensing, which provides a means to directly observe a few key variables. Satellite altimeters, for example, provide high resolution water level data of lakes and reservoirs (Schwatke et al., 2015; Busker et al., 2019; Biswas et al., 2019), while optical satellite images can be processed to map and detect changes in water surface area (Pekel et al., 2016; Zhao and Gao, 2018; Pickens et al., 2020). Moreover, data on water level and area can be combined with information on bathymetry (e.g., elevation-area curve) to infer the storage time series (see the review by Gao (2015)). The widespread availability of satellite data has indeed sparked research on monitoring of reservoir operations in several ungauged basins across the globe (Gao et al., 2012; Duan and Bastiaanssen, 2013; Bonnema et al., 2016; Busker et al., 2019), including the Mekong River Basin. For example, Liu et al. (2016) used satellite radar altimetry and Landsat images to estimate the water level of two reservoirs in the Lancang (Xiaowan and Jinghong) for the period 2000-2015. Their analysis was limited to cloudless Landsat images, so the time series so-derived have an irregular temporal resolution. Shortly after, Bonnema and Hossain (2017, 2019) estimated reservoir storage change for several sites of the Mekong, focusing primarily on its lower reaches.

Importantly, the aforementioned approaches and data have started to find their way into decision support systems used by the Lower Mekong countries. A first example is the Mekong Dam Monitor, an online platform for near-real time monitoring of dams developed by the Stimson Center and Eyes on Earth (https://www.stimson.org/project/mekong-dam-monitor/). Specif-

ically, the platform uses Sentinel 1 and 2 images to provide weekly updates of water level in the thirteen dams built on the main stem—plus fourteenth additional reservoirs on the river tributaries (Eyler et al., 2020). Because Sentinel 1 and 2 were launched in April 2014 and June 2015, respectively, the available time series are relatively short and do not include the filling period of the two largest Lancang's reservoirs, Nuozhadu and Xiaowan (for additional details on the difference between the methodology used in this study and the one adopted by the Mekong Dam Monitor, please refer to Text S1). Another example is the Reservoir Assessment Tool (RAT, https://depts.washington.edu/saswe/rat_beta/), an online tool for near real-time monitoring and impact analysis of existing and planned reservoirs (Biswas et al., 2021). RAT uses Landsat 5 and 8 images to monitor ~1,500 reservoirs in South America, Africa, and Southeast Asia, including six in the Lancang River Basin.

Notwithstanding these recent advances, a deeper understanding of dam operations in the Lancang River Basin is needed to inform the downstream countries and seek cooperative solutions spanning across the entire basin. A first complexity is the lack of water level and storage time series (for each reservoir in the Lancang Basin) with adequate temporal resolution and horizon—ideally, each time series should have at least a data point per month and cover the entire life span of a given dam. Here, an important challenge lies with data availability: Landsat images are available for almost any reservoir and span more than three decades, but are affected by clouds (Busker et al., 2019; Biswas et al., 2021), thereby requiring an image enhancement process (Gao et al., 2012; Zhang et al., 2014; Avisse et al., 2017). Conversely, satellite altimeter observations are less subject to external disturbances. However, they either have sparse spatial coverage (satellite radar altimeters)—data are not available for all reservoirs due to their narrow ground track and orbit—or have a long revisit time (satellite laser altimeters). The ICESat series (satellite laser altimeters), for example, has a 91-day return period. Second, we need to discover the filling strategy of these dams, that is, the rate with which they have been filled. Unveiling these strategies could help understand past changes in downstream water availability and prepare contingency plans, since China is planning to build ten more dams in the Lancang (MRC, 2020b). Third, the availability of monthly storage data is the prerequisite for any event attribution analysis on droughts and pluvials. In other words, detailed information on the operations of the Lancang's dams could help us explain whether or how they contributed to recent extreme events (Keovilignavong et al., 2021).

In this study, we address the three knowledge gaps described above. To this purpose, we rely on a 30 m Digital Elevation Model (DEM) from the Shuttle Radar Topography Mission (SRTM), satellite imagery (Landsat 5, 7, and 8) and altimetry water level data (Jason and Sentinel-3) (Section 2). In particular, we use the DEM data to identify the elevation-storage and area-storage curves and process the Landsat images to generate monthly time series of water surface area for each reservoir. In this analysis, we improve the algorithm introduced by Gao et al. (2012)—and modified by Zhang et al. (2014)—for processing cloudy images and tailor it to Landsat data. We then infer the time series of reservoir storage by combining information on water surface area and area-storage curve, and validate the results using the altimetry water level data with the elevation-storage curve (Section 3). With the storage time series at hand, we unveil the filling strategies, infer the rule curves, and relate the downstream hydrological alterations to the reservoir management strategies (Section 4). Building on this knowledge, we identify and discuss opportunities for improving the management of the Lower Mekong resources under present and future scenarios (Section 5 and 6).

## 2 Study Site and Data

### 2.1 Study Site

The Mekong is a transboundary river flowing across Southwest China and Southeast Asia (Figure 1(a)). The river originates from the Tibetan Plateau at an altitude of about 5200 m a.s.l. and flows in a northwest-southeast direction through six countries (China, Myanmar, Laos, Thailand, Cambodia, and Vietnam) before pouring into the East Vietnam Sea. The Mekong drains an area of 795,000 km$^2$ with an average annual discharge of approximately 475 km$^3$. Its upper portion is 2140-km long and drains an area of 176,400 km$^2$. The high mountains and low valleys characterizing the Lancang River Basin contribute to the spatial variability of precipitation, whose annual average varies from 750 to 1025 mm across the basin. Precipitation is also unevenly distributed across the year, with two distinct dry (December to May) and wet (June to November) seasons. The streamflow reflects a similar seasonal pattern (Yun et al., 2020). Although the drainage area of the Lancang River accounts for about 22% of the total catchment area, the Lancang contributes only to 16% of the average annual discharge of the whole Mekong River (MRC, 2009).

The advantageous topography and abundant water availability make the Lancang River Basin an ideal spot for the hydropower industry (Dang et al., 2020a). The first dam on the mainstream of the Lancang (Manwan) began its operations in 1992, followed by Dachaoshan in 2003 and Jinghong in 2008. The two largest dams (Xiaowan and Nuozhadu) became operational in 2009 and 2013, respectively. And since 2016, at least one dam joined the Lancang's reservoir system every year. Overall, this rapid transformation of the basin resulted in a system comprising eleven operational and one planned dam (Figure 1(b)).

The design of the cascade reservoir system reflects the topographic characteristics of the basin. Specifically, the presence of narrow valleys with steep sides required the construction of high dams (see Figure 2 and the list of design specifications in Table S1). In turn, this resulted in reservoirs with large storage capacity relative to inflow, steep banks, and long and horizontally narrow shapes. The total storage capacity is 42,170 Mm$^3$, about 55% of the average annual discharge at Chiang Saen gauging station, the first downstream station with publicly-available data (Figure 1). These reservoirs form a long and complex cascade system, so it is only by studying it in its entirety that we can understand how storage operating patterns has evolved over the past decade.

### 2.2 Data

In this study, we focus on the ten largest operational reservoirs (each with a volume larger than 100 Mm$^3$), all located on the main stem of the Lancang River. We select 2008–2020 as our study period because it includes the year of commission of most dams (eight out of ten); a choice that allows us to study their operations during the filling period as well as under regular operating conditions. Extending the temporal horizon to include the year of commission of the two remaining dams (Manwan and Dachaoshan, commissioned in 1992 and 2003) would complicate the analysis unnecessarily, since their aggregated storage capacity corresponds to only 2.14% of the current total system capacity. For the aforementioned study period we gathered data on Digital Elevation Model (DEM), satellite imagery, and radar altimetry water level.

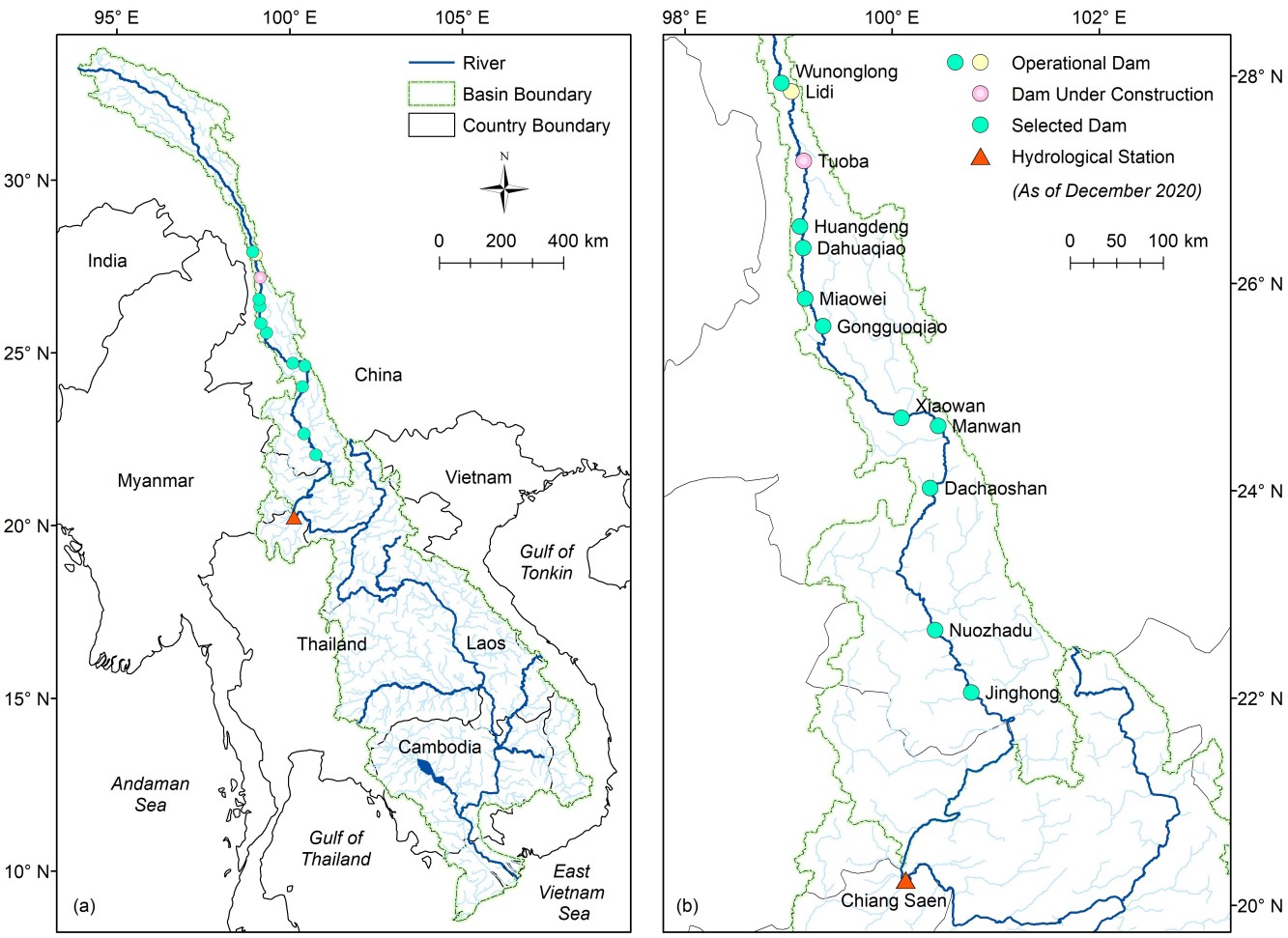

**Figure 1.** Mekong and Lancang River Basins ((a) and (b), respectively). In both maps we report the location of the gauging station as well as the hydropower dams on the main stem of the Lancang. All dams were operational as of December 2020, with the exception of Tuoba, which is currently under construction. The dams analyzed in our study are denoted by a green circle.

### 2.2.1 Digital elevation model

Digital elevation models contain the information on terrain elevation needed to represent reservoir bathymetry, so they are
commonly used to establish the relationship between water level and water surface area (Bonnema et al., 2016; Zhang and Gao, 2020). In this study, we use the global 30-m spatial resolution DEM obtained by the Shuttle Radar and Topography Mission (SRTM). The SRTM-DEM provides the terrain elevation above the water level at the observation time of the SRTM mission (February 2000) in signed integer raster format. The SRTM-DEM is the best choice for representing reservoir bathymetry on the Lancang River because of its high spatial resolution, acquisition time (nine out of ten selected reservoirs were constructed
after February 2000), and free accessibility. We note that the reservoir construction may have slightly changed the bathymetry,

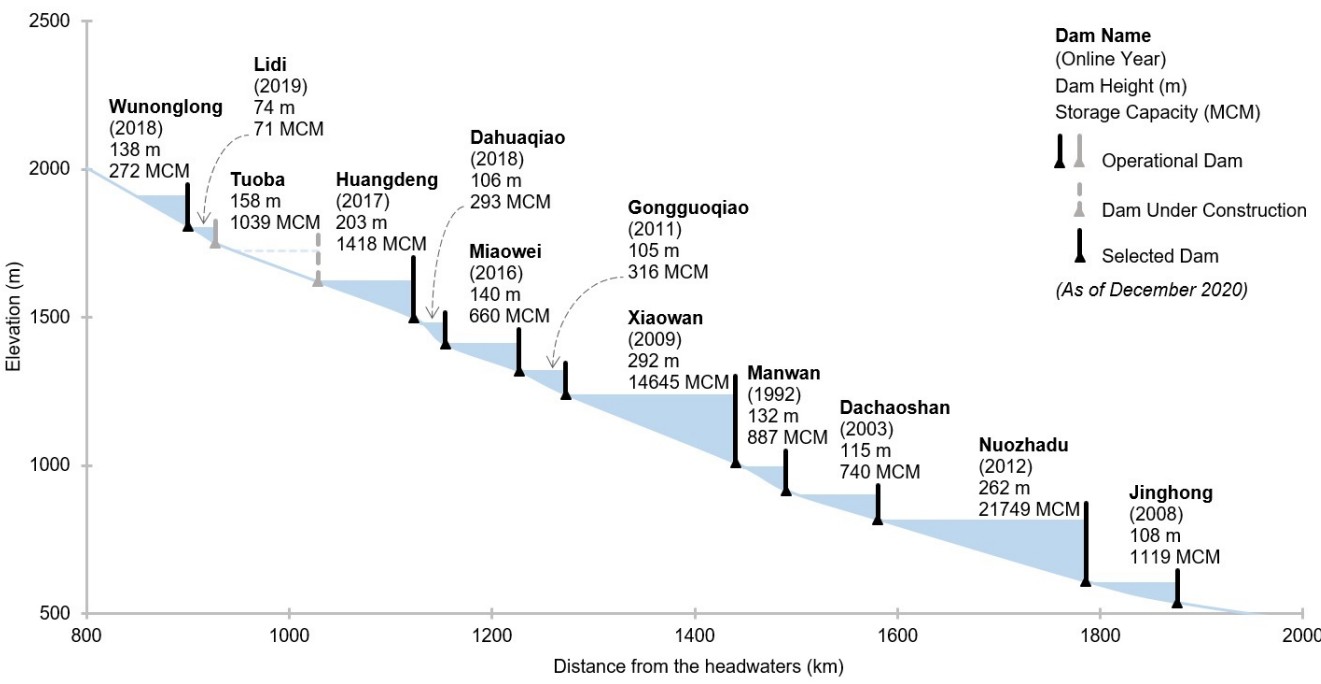

**Figure 2.** Cascade reservoir system on the Lancang River. Further details about the design specifications are provided in Table S1.

but these changes are negligible for our study site. That is because of two reasons. First, Lancang's reservoirs have horizontally narrow and long shapes. Their length varies from about 25 km (Dahuaqiao) to about 198 km (see Figure 2). Because of these characteristics, dam construction sites (typically carried out near the dam location) only affect a very small portion of the reservoir bathymetry. Second, Lancang's reservoirs have a large portion of dead storage, from about 32% (Xiaowan) to 87% (Wunonglong). Therefore, the reservoir bathymetry in the variation range of the reservoirs is barely affected by dam constructions.

### 2.2.2 Satellite Imagery

We use images from Landsat 5, 7, and 8 to estimate the water surface area of the Lancang reservoirs. That is because of four reasons. First, Landsat imagery has been collected for a long time, so it covers our study period. Second, Landsat images have a high spatial resolution (30 m), which is suitable to detect changes in the water surface area of reservoirs with long and horizontally narrow shapes, like the ones in our study site. For instance, the width (at full capacity) of Nuozhadu and Xiaowan reservoirs, the two largest reservoirs on the Lancang River, is only ~1500 and ~1000 m. Third, the frequency of Landsat imagery (16 days) is enough to assess the change of reservoir water surface area with a monthly time step—a reasonable temporal resolution for reservoirs characterized by massive storage capacities (see Figure S1). Moreover, we can double the number of images for each month, because the active period of Landsat 7 (1999–present) overlaps with the active period of

Landsat 5 (1984–2013) and Landsat 8 (2013–present). Fourth, Landsat imagery has been successfully used in other studies to estimate reservoir water surface area (e.g., Duan and Bastiaanssen (2013), Avisse et al. (2017), Bonnema and Hossain (2017)). The images used in this study are archived from the Landsat Collection-1 Level-2 (Surface Reflectance) of the United States Geological Survey (USGS). It is also worth mentioning here that (publicly available) imagery provided by other missions, such as MODIS (Moderate Resolution Imaging Spectroradiometer) and Sentinel, may not be best suited for this study. MODIS imagery has high frequency (twice a day) but lower spatial resolution (250 m), which makes it unsuitable for estimating the water surface area of medium and small reservoirs or large, but horizontally narrow, reservoirs (the width of all reservoirs, except for the two largest ones, varies from 300 to 600 m). Meanwhile, Sentinel has been operational since 2015, so its temporal coverage is not sufficiently long for our analysis. Further details concerning a comparison between Landsat, MODIS, and Sentinel imagery are reported in Table S2.

### 2.2.3   Radar Altimetry Water Level Data

Satellite radar altimeters have been used for decades to monitor the ocean and large reservoirs and lakes (Schwatke et al., 2015) —see Table S3 for additional details on satellite altimeters. Because radar altimetry data from each satellite are not available for all reservoirs, we make use of all available sources of radar altimetry data previously processed into water level time series— following the methods developed by the NASA Ocean Altimeter Pathfinder Project—and published by the Global Reservoirs and Lakes Monitor (G-REALM) (Birkett et al., 2010). Specifically, we use Jason-2 (2008-2016) for Nuozhadu, Xiaowan, and Huangdeng, Jason-3 (2016-2020) for Xiaowan and Huangdeng, Sentinel-3A (2016-2018) for Nuozhado, and Sentinel-3B (2019-2020) for Jinghong. As we shall see, the lack of radar altimetry water level data for the remaining reservoirs does not affect the conclusions of our study, since we use them only for the purpose of validating the results obtained through satellite imagery.

### 3   Methodology

Our methodology is chiefly aimed to estimate (and validate) the storage time series of each reservoir. To this purpose, we follow three main steps, illustrated in Figure 3. We begin by processing the information contained in the DEM to estimate the relationship between water level (WL) and water surface area (WSA) for each reservoir. With this relationship, also called the elevation-area (E-A) curve, we calculate the elevation-storage (E-S) curve (the relationship between WL and storage volume) and the area-storage (A-S) curve (the relationship between WSA and storage volume). Then, we estimate the WSA of each reservoir from all Landsat images available for our study period. To carry out this step, we rely on a novel variant of the WSA estimation algorithm developed by Gao et al. (2012) and modified by Zhang et al. (2014). Finally, we use the A-S curves and WSA time series to infer how the storage of each reservoir varied during the study period. We validate our methodology on two reservoirs located outside of the Lancang Basin: Bhumibol reservoir in Chao Phraya River Basin and Ubol Ratana reservoir in Lower Mekong River Basin, for which storage and water level data are publicly available (see Figure S2 and S3). A detailed

explanation of these steps is provided in Section 3.1 and 3.2. In Section 3.3 and 3.4, we describe the numerical approaches adopted to estimate the reservoir filling strategies and analyze the effect of reservoir operations on downstream discharge.

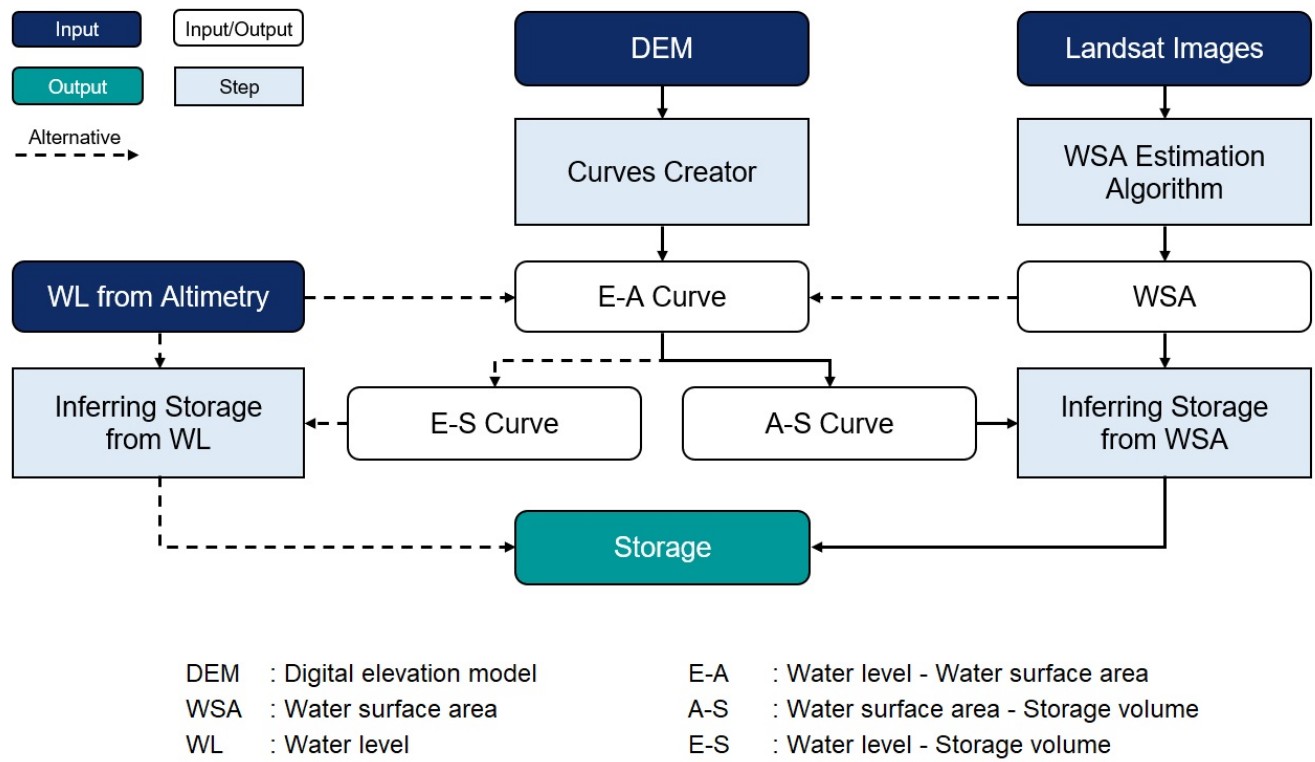

**Figure 3.** Flowchart representing our methodological approach. The two key steps are the calculation of of the E-A, E-S, and A-S curves (from the DEM) and the estimation of the WSA (from Landsat imagery). With this information at hand, we estimate the storage time series of each reservoir. The altimetry water level data are coupled with the E-S curve to re-estimate the storage time series with independent data, thereby validating the estimation based on Landsat imagery.

## 3.1 Estimating the E-A, A-S, and E-S curves

Recall that for nine, out of ten, reservoirs, the SRTM-DEM can provide full information on bathymetry (Section 2.2). To estimate the E-A curve of these reservoirs, we first isolate the DEM data with the contour corresponding to maximum water level and dam crest line. The purpose of this step is to calculate the curve within the extent of the reservoir only and thus avoid errors due to the inclusion of surrounding areas. Then, we calculate the surface area corresponding to each 1-m elevation of the DEM. Specifically, with each elevation value (each meter) from the lowest elevation within the reservoir extent to the maximum

water level, we count the number of pixels having a value equal to or smaller than that elevation value. This is because, when water reaches that elevation, the area corresponding to those pixels is inundated. Then, we multiply the number of pixels by the pixel size (30 m x 30 m) to get the water surface area. We finally fit a five-degree polynomial (degree determined by

trial-and-error) to the data points so obtained. For the remaining reservoir, Manwan, we apply the same procedure, but only to the portion above the water level recorded by the SRTM. To approximate the E-A curve below that water level, we fit a five-degree polynomial to the part above the water surface and then extend it below the water surface, as in Bonnema et al. (2016); Bonnema and Hossain (2017).

With the E-A curve at hand, we calculate the storage volume corresponding to each 1-m elevation of the DEM. This operation is carried out using the following trapezoidal approximation (Gao et al., 2012; Bonnema and Hossain, 2019; Li et al., 2019; Tortini et al., 2020):

$$V_i = \sum_{j=l+1}^{i} (A_j + A_{j-1})(E_j - E_{j-1})/2, \tag{1}$$

where $V_i$ is the storage volume corresponding to the water level $E_i$ and water surface area $A_i$, while $l$ denotes the lowest elevation of the reservoir bathymetry (i.e., $A_l = 0$).The trapezoidal approximation is used here instead of a direct calculation from the DEM because the latter is not applicable to Manwan—while it is desirable to minimize the differences in data processing for all reservoirs. Besides, with the E-A curves validated by water level observations (from altimetry data) and water surface area (from Landsat images), we can confidently develop the E-S and A-S curves from the E-A curves using the trapezoidal approximation (recall we do not have observed storage data to validate the E-S and A-S curves estimated directly from the DEM). To strengthen the rationale for using the trapezoidal approximation, we compare the results obtained with the two methods. The differences in storage corresponding to each water level in the variation range are not more than 1% (for Jinghong, Miaowei, Huangdeng, and Wunonglong) and 2% (for Nuozhadu, Dachaoshan, Xiaowan, Gongguoqiao, and Dahuaqiao). The detailed comparisons for Nuozhadu and Xiaowan reservoirs can be found in Table S4 and Figure S4. Finally, we use the data points on storage volume to fit the A-S and E-S curves. All aforementioned operations are carried out in Python 3.7 with the aid of the *OSGeo* library.

## 3.2   Inferring the water surface area

Water surface data can be inferred from Landsat images by classifying each pixel with either a single spectral band (e.g., near-infrared band) or a spectral index calculated from multiple bands (see Table S5 for a list of the most common indices). In general, the use of a single spectral band reduces the computational requirements (Li et al., 2019), but spectral indices tend to provide more robust results (Liu et al., 2016). Whatever the method used, one key challenge with Landsat images stands in the presence of clouds, cloud shadow, and no-data pixels (for Landsat 7), which may lead to a misclassification of water pixels and the consequent underestimation of the water surface area. To handle this problem, we use a novel variant of the WSA estimation algorithm introduced by Gao et al. (2012) and Zhang et al. (2014), originally conceived to extract water surface area from the Normalized Difference Vegetation Index (NDVI) layer—which is included in the 250 m-resolution global Terra MODIS Vegetation Indices (MOD13Q1), a level-3 MODIS product provided by NASA.

Like the modified version by Zhang et al. (2014), our algorithm consists of two main phases: mask creation and water classification improvement, illustrated in Figure 4 with light blue and light green boxes. In the first phase, the cloudless images are processed together to create two products, the expanded mask and zone mask. The two masks are then used in the second

phase, where the Landsat images are individually processed to obtain the water surface area corresponding to the collection time of each image. The major modifications with respect to the version by Zhang et al. (2014) are the selection of cloudless images (Step 1.1) and identification of additional water zones (Step 2.5); two modifications needed to ensure that the algorithm performs well with Landsat images (instead of the NDVI layer of MOD13Q1). Further details for each phase and step are provided below.

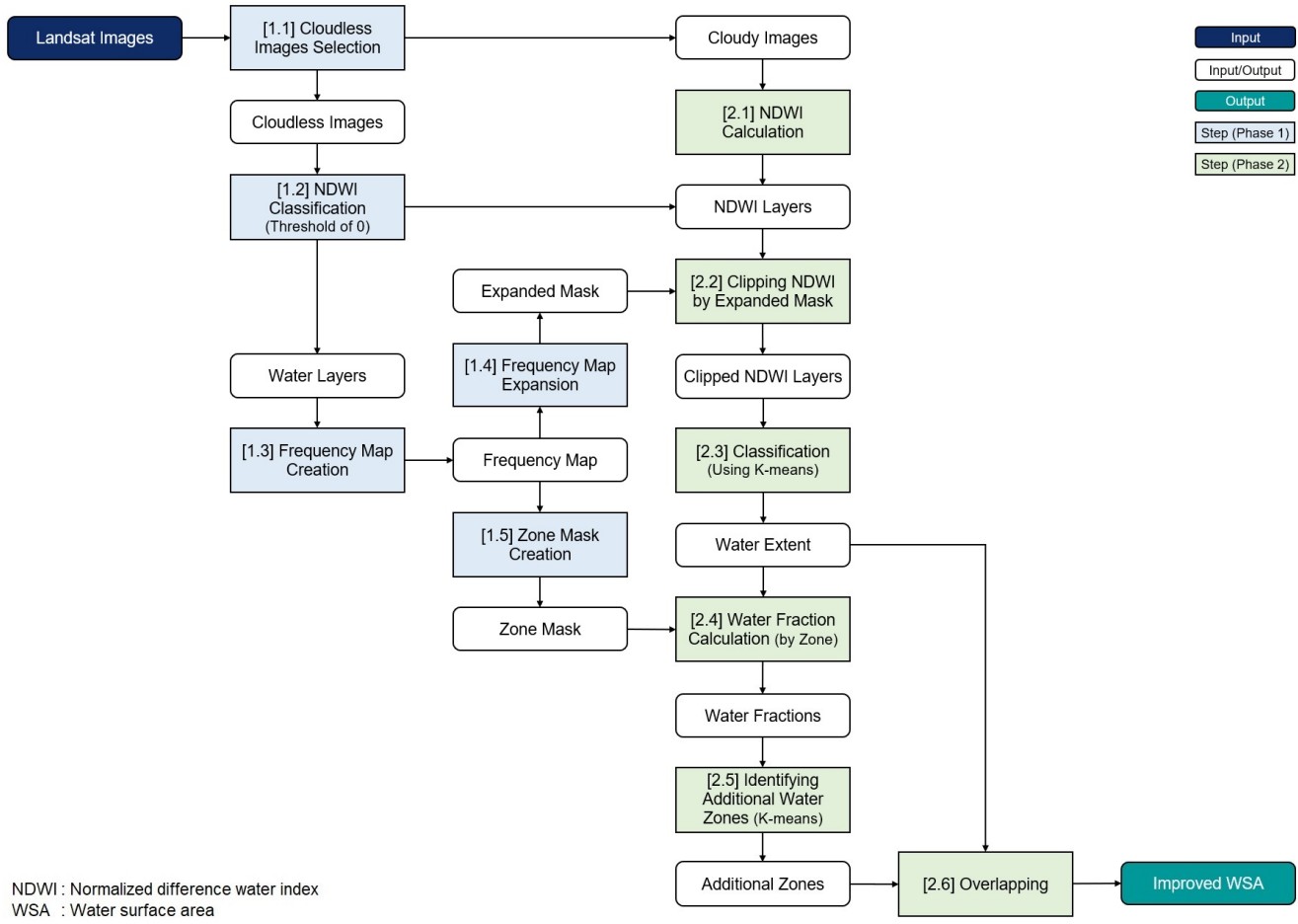

**Figure 4.** WSA estimation algorithm. The first phase is aimed at the creation of the expanded mask and zone mask, while the second phase focuses on the processing of each image to yield the water surface area.

*[1.1] Selection of cloudless images.* Cloudless images are the ones that do not contain clouds or contain very little clouds on the reservoir surface extent. For our application, we define a cloudless image as an image with less than 20% of cloud cover on the maximum reservoir surface extent. To identify these images, we use the BQA band (the band of quality assessment), which contains the information on cloud pixels. As we shall see, working on a subset of cloudless Landsat images is necessary to preserve the quality of the frequency map and masks produced in the next steps. Note that the version by Zhang et al. (2014)

did not include this step because cloud effects are partially removed from the NDVI layer in MOD13Q1 (Didan and Munoz, 2019). This is the result of selecting the best available pixel value (the low clouds and the highest NDVI value) from all daily acquisitions within a 16-day period.

*[1.2] NDWI-based classification.* To classify the water and non-water pixels, we use the normalized difference water index (NDWI) with a threshold value equal to 0. The choice of index and corresponding threshold is based on a preliminary analysis, in which we compared the performance of NDWI, NDVI, and MNDWI (Modified Normalized Difference Water Index) for all ten reservoirs. The results, reported in Figure S5-S9 for 60 cloudless Landsat images for each reservoir, show that the NDWI-based classification matches the maximum water extent reported in the Maximum Water Extent dataset, developed by the European Commission's Joint Research Centre (Pekel et al., 2016). On the other hand, NDVI and MNDWI tend to provide less reliable results. As for the threshold value, 0.05 and 0.1 (for NDWI) tend to lead to an underestimation of the water pixels, since the total number of times a water pixel is correctly classified as water is less than 60. We also manually checked the obtained water layers with the true colour Landsat images for a number of images before making our decision of using NDWI. The NDWI layers so-calculated are subsequently used in Step 2.2.

*[1.3] Frequency map creation.* To create the frequency map, we first calculate the percentage of times in which a pixel is classified as water (based on its NDWI value) in all selected cloudless images. This operation is carried out for all pixels within the bounding box of the reservoir extent. Then, we create the frequency map by selecting the pixels with frequency larger than 0. This step is illustrated in Figure 5(a,b).

*[1.4] Frequency map expansion.* We expand the frequency map by buffering it with three additional pixels; in other words, we add three pixels around the peripheral water pixels (see Figure 5(a,b)). The expansion is aimed to ensure that no possible water pixels are missed out. This 90-m buffer around the nominal shoreline is deemed sufficient for our case study, since reservoirs in the Lancang are located in steep terrains, where the storage is controlled by elevation more than area. The expanded frequency map is used in Step 2.2 to clip the NDWI layer; hereafter, we refer to it as expanded mask.

*[1.5] Zone mask creation.* In the last step of Phase 1, we convert the frequency map into a 50-zone mask. As illustrated in Figure 5(c), the $i$-th zone contains the pixels classified as water with a frequency greater than $2 \cdot (i-1)\%$ and less than or equal to $2 \cdot i\%$ (with $i = 1, \ldots 50$). For example, Zone 1 contains the pixels classified as water from 0 to 2% of the time, while Zone 2 contains those classified as water from 2 to 4% of the time. At the end of this phase, we obtain the two inputs for the next phase, that is, the expanded mask and zone mask.

*[2.1] NDWI calculation.* Here, we calculate the NDWI index for the remaining Landsat images—with clouds, cloud shadow, and no-data pixels—and pass them to the next step in the form of a raster layer for each image. Note that the goal of this second phase is to improve the water surface classification of the images, so as to maximize the number of data points available for our study period, especially for the monsoon season when Landsat observations are heavily affected by clouds. Failing to improve the cloudy images can make the water surface area estimates inaccurate.

*[2.2] Clipping the NDWI layer by the expanded mask.* The NDWI raster layer obtained in Steps 1.2 and 2.1 is clipped by the expanded mask created in Step 1.4.

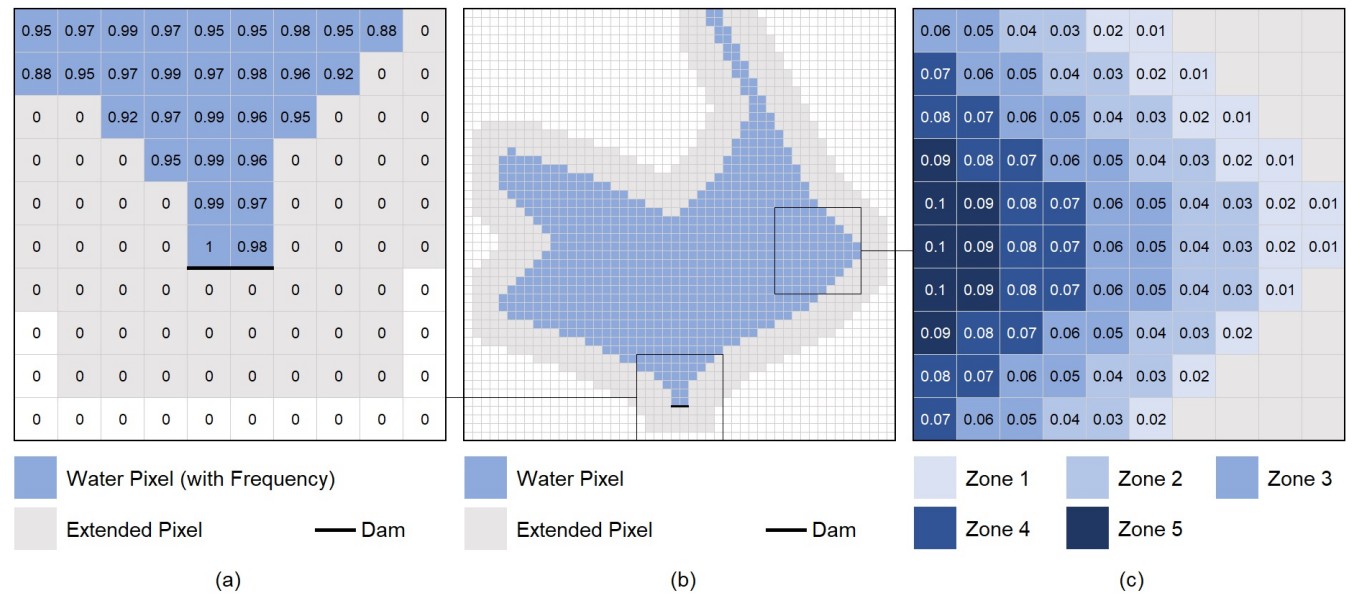

**Figure 5.** Example of a frequency map (a,b), expanded mask (a,b), and zone mask (c).

*[2.3] k-means-based classification of the water pixels.* Because of the presence of clouds, and other disturbances, the of use of the same NDWI threshold (equal to 0) in all Landsat images may lead to overestimation or underestimation errors of the water surface area. To find NDWI thresholds for each Landsat image, we resort to $k$-means clustering. Specifically, we set $k$ equal to three (a value found by trial-and-error) and apply $k$-means clustering to all pixels in the NDWI layer (Figure 6(a)). Water pixels tend to fall into the cluster with the highest NDWI values, because the NDWI of water pixels has higher value than the one of non-water pixels. Results are verified by manually checking the classified water layer with true-color Landsat images.

*[2.4] Water fraction calculation (by zone).* The zone mask created in Step 1.5 is used here to divide the water extent layer (obtained in the previous step) into 50 zones. For the $i$-th zone, we define the water fraction $p_i$ as follows:

$$p_i = n_i/N_i, \ i = 1, 2, ..., 50, \tag{2}$$

where $p_i$ represents the ratio between the number $n_i$ of pixels classified as water in zone $i$ (with the NDWI-based $k$-means clustering) and the total number $N_i$ of pixels in zone $i$ (retrieved from the zone mask). The information provided by the water fraction of each zone is used in the next step to improve the water pixel classification.

*[2.5] Identification of additional water zones.* We improve the classification of water pixels by identifying the additional water zones based on their water fraction. To do so, we resort again to the $k$-means clustering algorithm. Moreover, because of the continuity of water extent (water expands from higher frequency to lower frequency zones), we also take into account the zone number (or frequency value). Then, we formulate a clustering problem in a two-dimensional space constituted by water fraction and zone number. We solve the clustering problem with a value of $k$ equal to two, found by trial-and-error. Figure

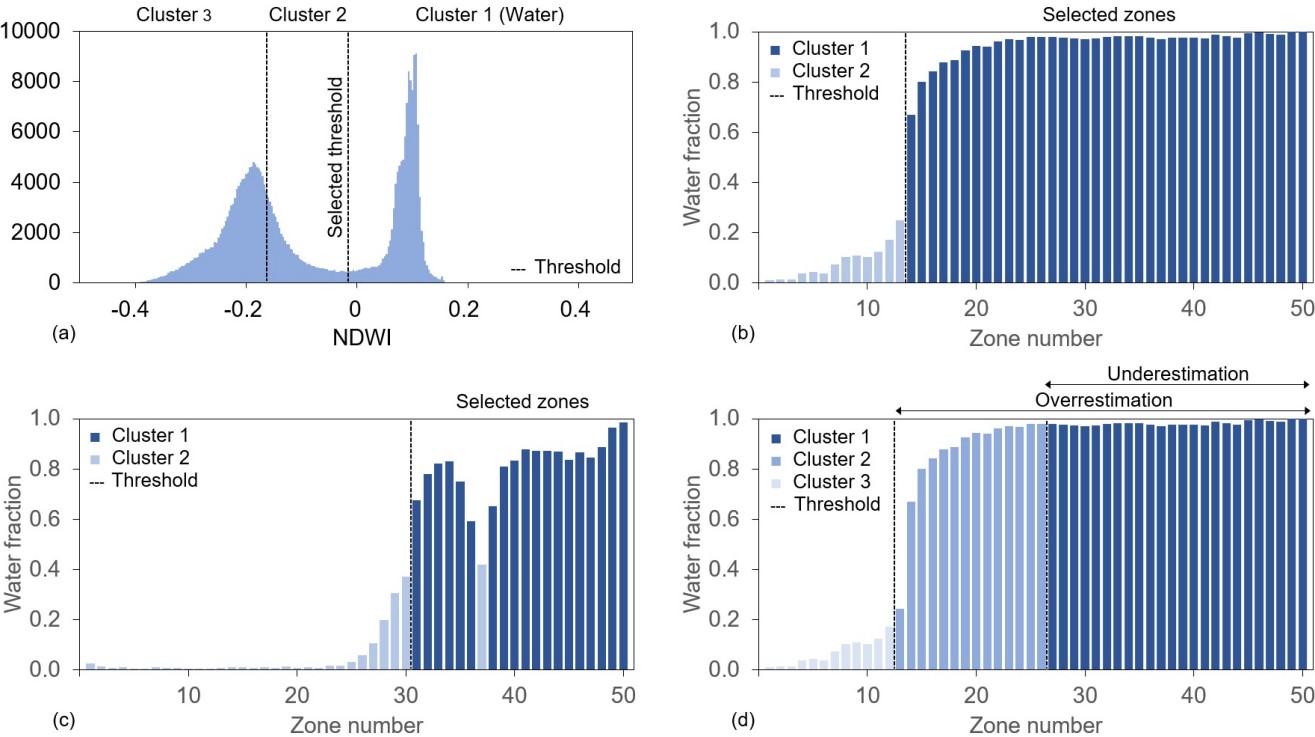

**Figure 6.** Illustration of the $k$-means classifications used in Step 2.3 and 2.5. Panel (a) shows the water pixels classification based on NDWI values (Step 2.3), while panels (b,c) show the identification of additional water zones based on two clusters (Step 2.5). Panel (d) illustrates the issues that raise when using three clusters in Step 2.5.

6(b,c) shows two examples with $k$=2, while Figure 6(d) reports an example for an unsuitable value of $k$. The lowest zone in the higher cluster (zone 14 in Figure 6(b) and zone 31 in Figure 6(c)) is the threshold above which zones are converted to water pixels. The reason for converting all pixels from the threshold zone onwards to water pixels is that the pixels in the same zone have the same (or very similar) inundation probability, and, at each observation time, they fall into one of two scenarios: (1) they are both non-water pixels or (2) they are both water pixels (even when the water fraction of that zone is less than 100%, due to cloud cover). Naturally, there can be a small error from the threshold zone. For example, Zone 14 in Figure 6(b) contains pixels with 26-28% inundation probability, but sometimes, the threshold of inundation probability is not exactly 26% (e.g., 26.5%, 27%, ...). Note that we could increase the performance by dividing the frequency map into a larger number of zones, but this would require a larger number of cloudless images. This step represents the second modification of the original WSA estimation algorithm, which uses a quality parameter not suitable for Landsat images—since the cloud effects are not mitigated, unlike the NDVI layer in MOD13Q1.

*[2.6] Overlapping.* Finally, the layer of additional water pixels is overlapped to the layer of water extent obtained in Step
2.3. The final output is the improved water classification for each image characterized by cloud cover or other disturbances.
All the aforementioned operations are carried out in Python 3.7 with the aid of the *OSGeo* and *SKLearn* libraries.

### 3.3 Inferring the reservoirs' filling strategies

Determining the filling strategy of a reservoir means deciding the rate with which the reservoir is filled and, therefore, the
fraction of inflow that is retained on a periodic basis—monthly, in our case. The problem is formalized by the following mass
balance equation:

$$S_t = S_{t-1} + \theta \cdot Q_t - E_t, \tag{3}$$

where $S_t$ is the reservoir storage at time $t$, $Q_t$ the inflow volume in the interval $(t-1, t]$, $E_t$ the evaporation loss in the interval
$(t-1, t]$, and $\theta$ a parameter varying between 0 and 1 and expressing the fraction of inflow volume retained by the reservoir. In
our case, the goal is to determine the value of $\theta$ (in each month) for Nuozhadu and Xiaowan during the periods 2012-2013 and
2009-2010, respectively.

Observed inflow data are not available, so we resort to modelled ones. Specifically, we use daily inflow data simulated by
VIC-Res (Dang et al., 2020a, b), a variant of the Variable Infiltration Capacity (VIC) model—a large-scale, semi-distributed
hydrological model first developed by Liang et al. (2014). Similarly to VIC, VIC-Res contains a rainfall-runoff module and a
routing module. In the first module, the study region is organized as computational cells with a customizable cell size (0.0625
degree in our case), in which key hydrological processes (evapotranspiration, infiltration, baseflow, and runoff) are calculated
as a function of hydro-meteorological forcings (precipitation, temperature, wind speed, etc.) and soil parameters. Then, the
routing module routes the simulated runoff and baseflow throughout the river network by using the linearized Saint-Venant
equation (Lohmann et al., 1996, 1998). In VIC-Res, the routing process includes a detailed representation of water reservoir
operations: for each reservoir in a given study site, the model calculates the mass balance and release, with the latter determined
by operating rules or pre-defined release time series. VIC-Res has been tested on several sites, including the Lancang River
Basin (over the period 1996–2005). In particular, Dang et al. (2020a) reported the results of model calibration with observed
discharge at Chiang Sean station and model validation with observed discharge at Jiuzhou station, located right upstream of
Xiaowan reservoir.

Here, we bank on the same VIC-Res model, which we force with rainfall and temperature (maximum and minimum) data
retrieved from CHIRPS-2.0 and ERA5 dataset. Land use and cover data were obtained from the Global Land Cover Char-
acterization dataset, while the soil data were extracted from the Harmonized World Soil Database. The monthly Leaf Area
Index and albedo were derived from the Terra MODIS satellite images, which provide changes in canopy and snow coverage
over time. The flow direction map used by VIC-Res is based on the SRTM-DEM. Since Dang et al. (2020a) considered the
dams built before 2005 and used the rule curves proposed by Piman et al. (2013), we slightly adapt the model to handle two
challenges for our current study: (1) consider more reservoirs (all dams on the mainstream built until 2013) and (2) leverage
the actual storage data retrieved from satellite data. To setup VIC-Res in our study, we therefore proceeded as follows. For

each reservoir, we take data on inflow (simulated), storage (estimated from the satellite data), and evaporation (simulated using the estimated water surface area and evaporation rates calculated with the Penman equation). We then invert the mass balance equation to calculate the release, which is used as input to VIC-Res to simulate the storage dynamics of each reservoir. The process is repeated sequentially—starting with the most upstream dam—so as to ensure that the cascading impacts of dams are captured correctly. To ensure the relaibility of this analysis, we extend the model validation at Chiang Sean for the filling period of Nuozhadu and Xiaowan (2009-2013); see Figure S10. The comparison between modelled and simulated storage is reported in Figure S11.

## 3.4 Indicators of hydrological alteration

The availability of storage data also allows us to decipher the impact of dam operations on downstream (measured) discharge. To do that, we calculate two time-varying indicators of hydrological alteration ($I_1$ and $I_2$). $I_1$ represents the fraction of the natural flow retained in the reservoir system for each month in which the system is storing water (i.e., when $\Delta S_t = S_t - S_{t-1}$ $> 0$). $I_2$ represents the fraction of the actual flow released from the reservoir system for each month in which the system is releasing water (i.e., $\Delta S_t < 0$). $I_1$ and $I_2$ are calculated as follows:

$$I_{1,t} = \frac{\Delta S_t}{\Delta S_t + Q_t}, \tag{4}$$

$$I_{2,t} = \frac{\Delta S_t}{Q_t}, \tag{5}$$

where $Q_t$ is the observed discharge volume downstream of the reservoir system (at Chiang Saen) in month $t$. Note that the denominator in eq. (4) approximates the natural flow in month $t$ (it is the sum of actual discharge and volume of water retained upstream in a given time interval).

## 4 Results

We begin this section by reporting the results of the analysis of DEM and satellite imagery, that is, the E-A, A-S, and E-S curves (Section 4.1) and water surface area (Section 4.2). We then present the storage time series of each reservoir, the information we use to retrieve the dam operating policies under filling and steady-state conditions (Section 4.3). Finally, we leverage these results to analyze the effect of reservoir operations on downstream discharge (Section 4.4).

## 4.1 E-A, A-S, and E-S curves

The E-A curves of Nuozhadu and Xiaowan reservoirs are illustrated in Figure 7 (panels (a) and (d)), where the blue circles represent the data points derived from the DEM, and the light blue lines are the five-degree polynomials fitted to them. Note that both curves correctly intersect the point identified by maximum water level and maximum water surface area, retrieved

from Do et al. (2020). A similar evaluation is carried out for the A-S and E-S curves (Figure 7, panels (b,c,e,f)), but this time using design specifications on full storage (A-S and E-S curves) and dead storage (E-S curves).

We carry out an additional validation of the E-A curves by comparing them against observations of water level and surface area obtained from radar alimetry data and Landsat imagery. These observations, illustrated in Figure 7 (a,d) by cyan diamonds, follow closely the curves identified through the DEM. Naturally, the cyan points are primarily concentrated between the dead and maximum water levels, which denote the normal range of operating conditions. As we shall see later, points below the dead water level correspond to the dam filling period.

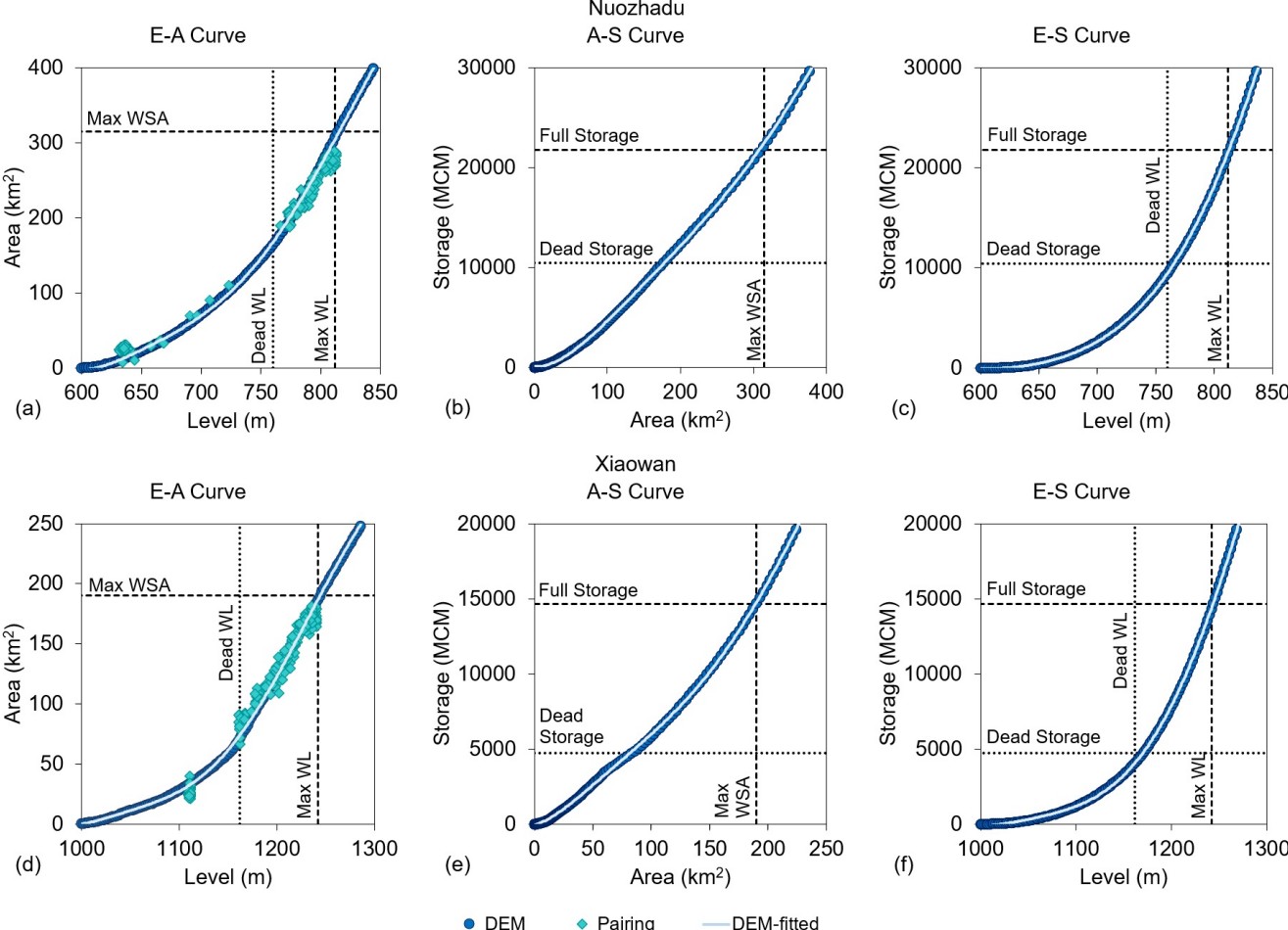

**Figure 7.** E-A, A-S, and E-S curves of Nuozhadu (top) and Xiaowan (bottom) reservoirs. The curves are represented by light blue lines, which are fitted to the data points (blue circles) derived from the DEM data. Note that the curves intersect the points identified by maximum water level, maximum water surface area, and full storage volume (dashed lines) as well as those identified by dead water level and dead storage volume (dotted lines). The cyan diamonds reported in panels (a) and (d) correspond to observations of water level and surface area obtained from from altimetry data and Landsat imagery.

The E-A, A-S, and E-S curves of the remaining eight reservoirs are reported in Figure S12 and S13. Apart from the curves of Jinghong and Huangdeng, which are evaluated by using radar altimetry water level, the curves of the other reservoirs can only be evaluated by comparing them against the design specifications reported by Do et al. (2020). Such evaluation is only partially successful, since we did not find a perfect match between curves and design specifications for Jinghong, Gongguoqiao, Miaowei, Dahuqiao, and Wunonglong reservoirs. Considering that the procedure used to estimate the curves has been successfully employed in several studies (Bonnema et al., 2016; Bonnema and Hossain, 2017; Zhang and Gao, 2020), we suspect that the reason behind the mismatch may lie with the information on dam design specifications available to the public. In turn, this reinforces the need for research aimed to retrieve data on large-scale infrastructure in transboundary river basins. We also note that this source of uncertainty does not severely affect our study, since those five reservoirs account for a small fraction of the total system's storage (2.36%, 0.74%, 1.55%, 0.69%, and 0.64%, respectively).

## 4.2 Water Surface Area

Recall that the WSA estimation algorithm builds on the idea of using cloudless images to create the expanded mask and zone mask, which are then employed to correct the classification of water pixels in images affected by clouds and other disturbances. In our case, such improvement is needed for 56% of the 3,004 Landsat images available for our study period (the number of usable images increases from 26% to 82%). As one might imagine, the classification correction is particularly important during the wet season, when cloud cover is more frequent—number of usable images increases by 54% of 1,770 images (from 30% to 84%) in the dry season and 58% of 1,234 images (from 21% to 79%) in the wet season. The performance of the algorithm for each reservoir is summarized in Table S6.

The WSA time series of Nuozhadu and Xiaowan reservoirs are reported in Figure 8. The first result to note is the stark change in the WSA values before (light blue points) and after (cyan points) the classification improvement. The time series of corrected WSA values also starts to reveal the reservoirs' operating patterns: the sharp increase beginning in 2012 (Nuozhadu) and 2009 (Xiaowan) denotes the starting point of the reservoir filling period, while the large, annual, fluctuations suggest the presence of a broad range of operating conditions—the maximum surface area is reached only at the end of the wet season, while the rest of year seems to be used to fill in and empty the reservoirs. In Section 4.3, we will see how such variability translates into storage patterns.

To evaluate the results obtained with Landsat imagery, we leverage the radar altimetry water level data and E-A curves to obtain two independent WSA time series. As shown in Figure 8, both modelling approaches provide very similar results. The same outcome can be seen in the WSA time series of Huangdeng and Jinghong reservoirs (see Figure S14). We also provide a quantitative comparison of Landsat-derived and altimetry-converted WSA for all four reservoirs mentioned above (see Table S7). With this additional analysis we therefore serve two purposes: scrutinize the WSA values for the main reservoirs and empirically validate the approach based on Landsat imagery, the only one available for the remaining reservoirs.

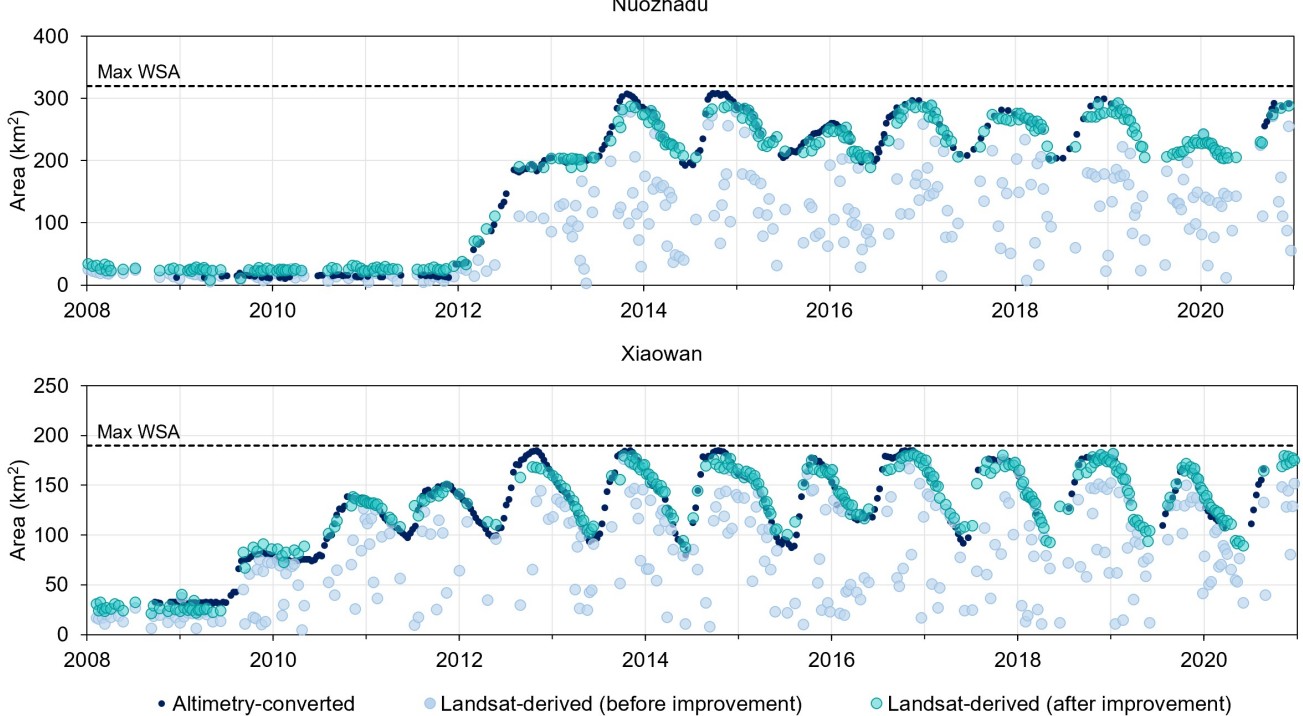

**Figure 8.** Water surface area of Nuozhadu (top) and Xiaowan (bottom) reservoirs. Note the drastic difference in WSA values before (light blue points) and after (cyan points) the classification improvement. The corrected values of WSA are well in agreement with those obtained through altimetry water level data and E-A curves (dark blue points).

## 4.3 Reservoir Storage

### 4.3.1 A history of reservoir storage variations

Using the information on reservoir curves and water surface area described above, we estimate the storage time series of each reservoir as well as their aggregated value (Figure 9). Note that the number of usable images per month is not the same. To have an evenly spaced time series of storage, we choose one WSA value (maximum value) for each month to infer the reservoir storage. The latter (dark blue line) portrays a history of rapid transitions, characterized by two major tipping points: the commission of Xiaowan and Nuozhadu reservoirs. After the commission of Xiaowan, we note a steady increase in the total

storage (see the period between mid 2009 to 2012); an increase that becomes even more pronounced after the commission of Nuozhadu, in 2012. It is indeed only after the filling of both reservoirs is completed, in 2014, that the total storage time series begins to exhibit a more cyclo-stationary behaviour—the reservoir system is filled during the monsoon season and emptied thereafter. The construction of a few additional dams during the period 2016–2018 does not seem to dramatically affect this pattern. In fact, the remaining eight reservoirs appear to maintain a more constant storage (Figure S15).

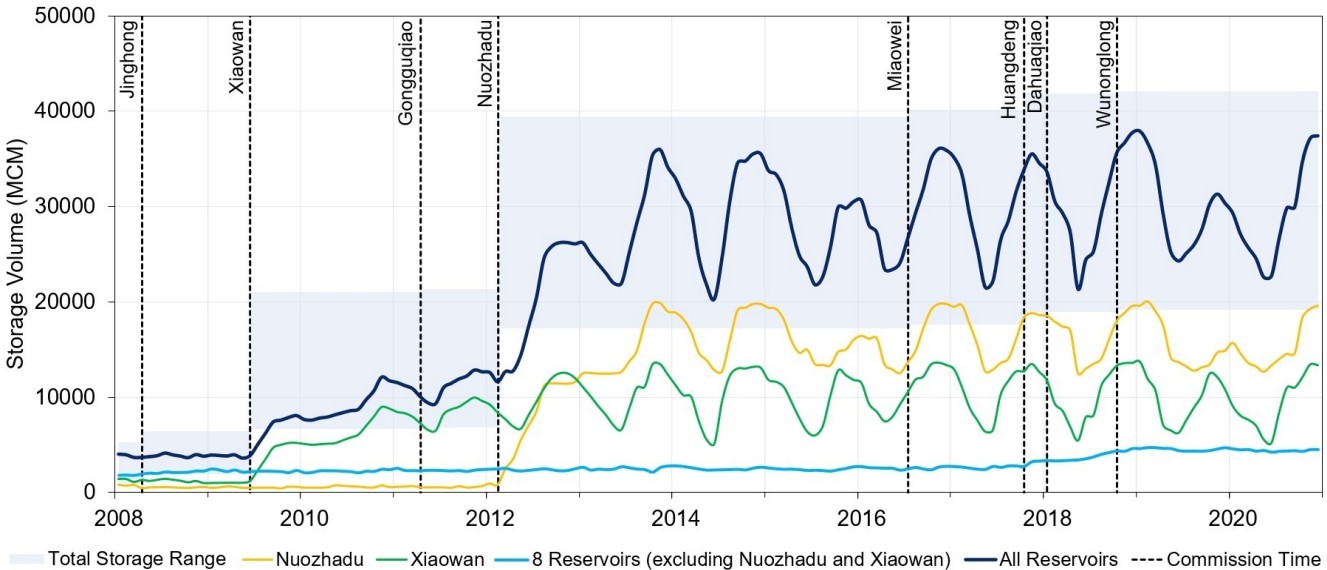

**Figure 9.** The blue shaded area represents the range of variability of the total system's storage (between dead and full storage volume), while the actual storage dynamics are represented by the dark blue line. The storage dynamics of Nuozhadu, Xiaowan, and the remaining eight reservoirs are illustrated by the yellow, green, and blue lines. The vertical dashed lines denote the year of commission of each reservoir. Note that Manwan and Dachaoshan began operations in 1992 and 2003, respectively. We provide the storage time series of each individual reservoir in Figure S15.

Two key additional elements are revealed when comparing the total storage dynamics against its potential range of variability, that is, the space between the aggregated dead and full storage (blue shaded area). First, the operators do not seem to use the entire storage at their disposal—dead and full storage levels were never reached throughout the study period. A plausible explanation for this management strategy may be sought in the need of avoiding further disputes with downstream countries (Eyler and Weatherby, 2020) or alleviating hydropower curtailment (Liu et al., 2018). Second, the reservoir system was used

at only half of its capacity in 2015-2016 and 2019-2020, with Nuozhadu reservoir playing a key role (yellow line). As we shall see in Section 4.4, this may be the result of persistent dry conditions (Yu et al., 2020; Ding and Gao, 2020), rather than a response to the aforementioned socio-technical drivers.

### 4.3.2  Filling strategies and operating rules

In Figure 10, we focus on the filling strategies and operating rules of Nuozhadu and Xiaowan: panels (a,b) show the values

of $\theta$ (the parameter expressing the fraction of inflow volume retained by the reservoir), while panels (c,d) illustrate storage volume (dark blue line), simulated inflow (green line), and storage change (light blue line)—that is, $S_t - S_{t-1}$, expressing the rate with which the reservoir is filled. The figure suggests that the operators adopted similar filling strategies: both reservoirs were filled in about two years (regardless of the different capacities), with the first wet season used to meet the dead storage

and the second wet season used to double the storage volume. Interestingly, results indicate that the annual value of $\theta$ was kept

constant during the filling period. For Nuozhadu, the operators retained 23% of the annual inflow volume (for both years); for Xiaowan, that value was kept to 17% and 15%. Note that these are extremely large values: retaining 23% of the annual inflow volume to Nuozhadu means storing roughly 9880 Mm$^3$, $\sim$12% of the average annual discharge at Chiang Saen. The filling strategy of the remaining reservoirs is different: because they have smaller storage capacity—relative to inflow—they are filled in a few months (see Figure S15).

By looking at the storage data of Nuozhadu and Xiaowan during normal operating conditions (i.e., once the filling is completed), we can get a few additional insights about the current management strategies (Figure 10 (e,f)). The first thing to note is the emergence of the seasonal patterns mentioned in the previous section; reservoirs are emptied during the pre-monsoon season and filled in thereafter. Second, the envelope of variability is rather broad, meaning that operators can deviate from the long-term pattern represented by the red bolded line. Such deviations are common throughout the entire Mekong Basin

(see Bonnema and Hossain (2017, 2019)) and are caused by inter-annual variability in discharge triggered by oceanic drivers (Nguyen et al., 2020). Finally, the analysis confirms that Nuozhadu and Xiaowan have not yet been used at their full capacity. However, this is is enough to keep the storage of the other reservoirs within a narrower range (Figure S16).

## 4.4    Impacts of Reservoir Operations on Downstream Discharge

Having established how the reservoirs in the Lancang River Basin have been filled in and operated, we can finally explain

their time-varying influence on the discharge measured at Chiang Saen (Section 2.1). The graphical analysis of total storage and discharge (Figure 11 (c)) highlights the stark changes in the flow regime in response to the increase in upstream storage. The flow regime changed drastically in late 2013, when the filling of Xiaowan and Nuozhadu was completed. By discharging water during the dry season and retaining it in the wet season, the hydropower dams largely increase low flows and decrease high flows (Table S8). For example, the mean of the annual peak discharge decreased from 11,157 (1990–2008) to 6,186 m$^3$/s

(2013–2020) (-45%), while the mean of the annual lowest discharge grew from 638 to 1,003 m$^3$/s (+57%). Similar figures are found for other statistics (Table S8). We can also note a macroscopic change in the seasonal discharge pattern, from ample annual fluctuations to more rapid flow changes. All these observations are confirmed by the wavelet analysis reported in Figure S17.

As shown in Figure 11 (b), the degree of flow alteration at Chiang Saen caused by the Lancang's dams increased significantly

over time with three distinct stages: the first stage (before Xiaowan reservoir began operating), the middle stage, and last stage (after Nuozhadu reservoir began operating). That means the range of variability of $I_1$ (in black color) and $I_2$ (in red color) increased over time; [0, 0.04] and [-0.10, 0] in the first stage, [0, 0.20] and [-0.44, 0] in the second stage, and finally, [0, 0.50] and [-0.91, 0] in the last stage. With the number of reservoirs increasing rapidly in the last decade, the downstream discharge became increasingly controlled by dam operations.

By bringing the monthly precipitation anomalies (for the Lancang River Basin) into the overall picture (Figure 11 (a)), we can better understand how dam operations partially contributed to downstream droughts and pluvials. A case in point is the drought in the period 2019–2020. The monthly precipitation anomalies show that, in the wet season of 2019, the Lancang

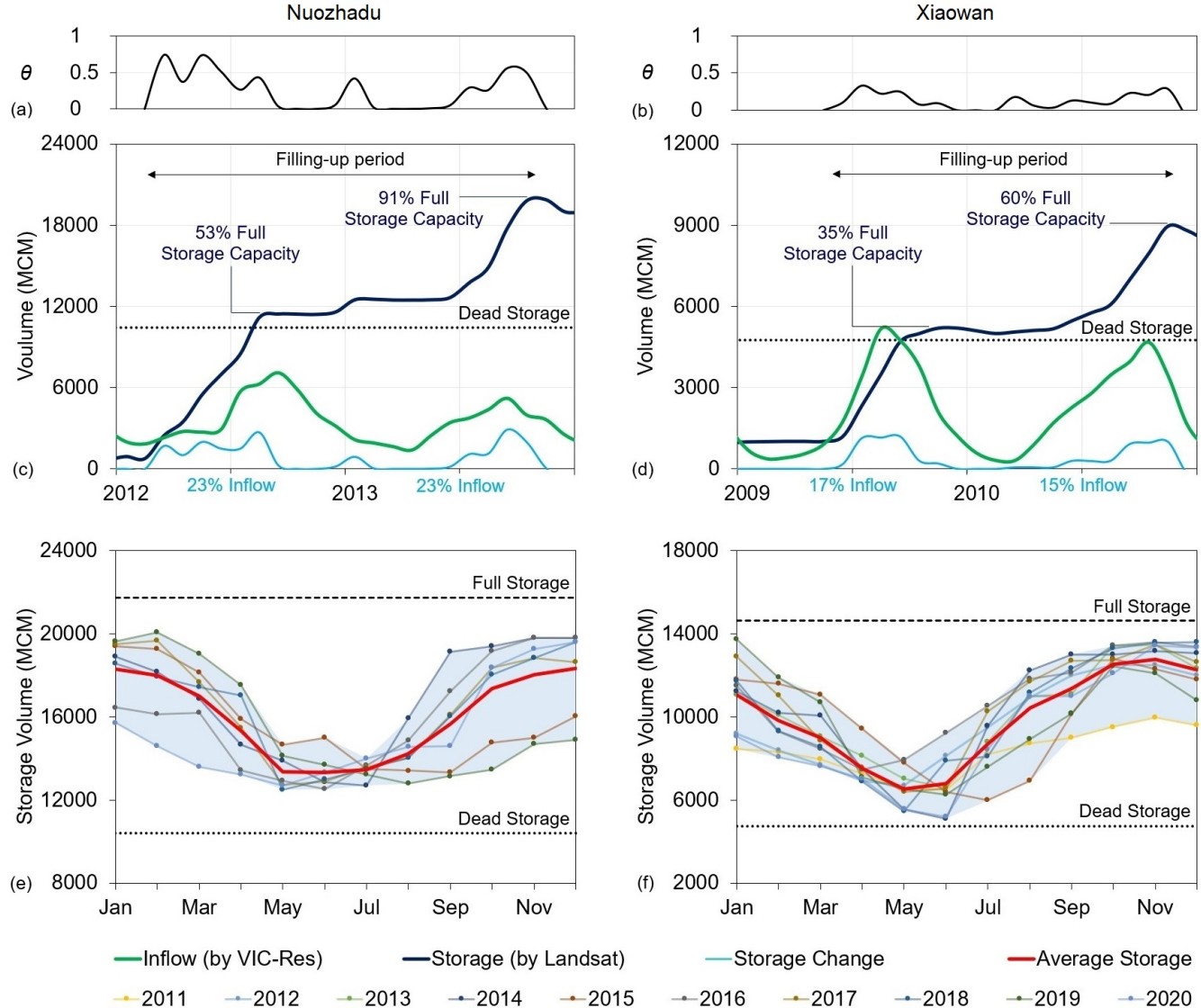

**Figure 10.** Filling strategies (a,b,c,d) and rule curves (e,f) of Nuozhadu (left) and Xiaowan (right) reservoirs. Panels (a,b) show the values of $\theta$. In panels (c,d), the storage volume (dark blue line) is derived from DEM and Landsat data, while the inflow to the reservoir (green line) is calculated with the VIC-Res hydrological model. The storage change (light blue line) is defined as the difference in storage volume between two consecutive months. In panels (e,f), each line with circle makers illustrates the storage volume of a given year. The red bolded lines represent the average monthly storage volume, considered representative of the rule curves. All data visualized here have a monthly resolution.

River Basin received less precipitation, especially in May and June (around 50 mm less than the average for those months). However, the values of $I_1$ indicate that the reservoir system kept retaining part of the inflow during the central months of the

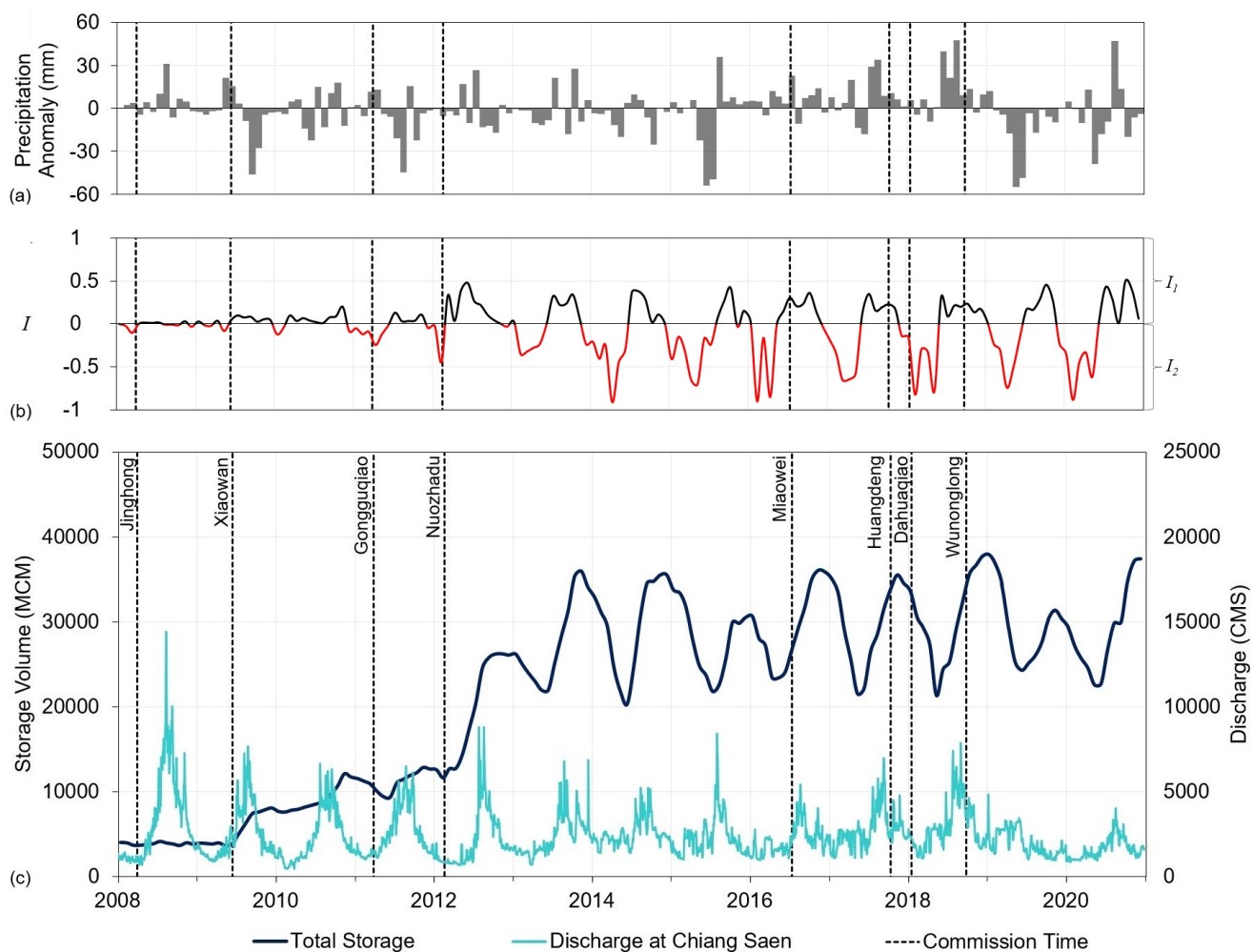

**Figure 11.** Impacts of reservoir operations on downstream discharge. Panel (a) shows the monthly precipitation anomaly in the Lancang River Basin, calculated from the CHIRPS-2.0 dataset. Panel (b) represents the two indicators of hydrological alteration on discharge at Chiang Sean: $I_1$—the fraction of the natural flow retained in the reservoir system for each month in which the system is storing water—in black color, and $I_2$—the fraction of the actual flow released from the the reservoir system for each month in which the system is releasing water—in red color. In panel (c), the bolded dark blue line represents the total storage of the reservoir system, while the cyan line represents the observed discharge at Chiang Saen.

year (up to about 46% in October). Because of such combination of meteorological drought and dam operating strategies, the downstream area underwent a critical dry period, with Chiang Saen gauging station recording extremely low flows during the summer months (MRC, 2020a). The release of water during the subsequent dry season only partially alleviated the effect of the ongoing drought, since the negative precipitation anomaly persisted until mid-2020. Importantly, the 2019–2020 data suggest that the dam operating strategy was not largely affected by the meteorological conditions: the Lancang dams currently

store about 46% of the estimated natural flow during the wet season (regardless of the monsoon's intensity) and then discharge it during the dry one, controlling up to 89% of the dry season flow—a pattern that emerged since Xiaowan and Nuozhadu became fully operational. These results also highlight the importance of emergency releases from the upstream reservoirs, such as those that were implemented in 2016 (Tiezzi, 2016; Hecht et al., 2019). In regard to those emergency releases, it should be noted that the 2016 drought had much smaller magnitude than the 2019-2020 one and that the drought occurred during the first half of the year, when the reservoir system was releasing water following its normal operations. In sum, the availability of (inferred) storage data can help us put droughts, emergency releases, and pluvials into a broader perspective. Yet, such analyses should ideally be complemented by more significant data sharing efforts among all riparian countries, a point that was recently reinforced by Keovilignavong et al. (2021).

## 5 Discussion

Our study produced a monthly storage time series for each of the ten large reservoirs built in the Lancang River Basin during the past decades. These time series describe the evolution of a massive dam cascade system and highlight the pivotal role played by Xiaowan and Nuozhadu reservoirs. Taken together, the two reservoirs can make up to ∼85% of the total system's storage in the Lancang, therefore largely controlling water availability in Northern Thailand and Laos. Bespoke information on their operating rules—ideally combined with real-time storage monitoring—is of paramount importance for many downstream socio-economic sectors. Consider, for instance, the Laotian hydropower industry, the largest regional exporter of electricity: since the construction of Xayaburi dam (1285 MW) on the main stem of the Mekong, part of the national hydropower production depends on the state of the Lancang's reservoirs. Detailed information on their storage and operating rules could therefore be incorporated into Laos' energy system models (Chowdhury et al., 2020), so as to address the asymmetric relation between China and Laos. Moving downstream, another sector that could benefit of our study are the Mekong's wetlands, a major biodiversity hotspot that is home to a multi-billion dollar fishing industry (Arias et al., 2014; Dang et al., 2016). Again, information on the state of the Lancang's reservoirs could help inform the operations of the many downstream dams, thereby helping implement release strategies that are less harmful for the environment (Sabo et al., 2017). In sum, the inferred rule curves could be used to predict outflow from the Lancang's reservoir system and adapt the operations of downstream dams.

Our analysis also provides a detailed description of the filling strategy of Nuozhadu and Xiaowan. We now know that both reservoirs reached steady-state operations in about two years by retaining from 15% to 23% of the annual inflow volume. This information is necessary to explain past anomalies in downstream water discharge and, most important, to prepare for future infrastructural changes in the Lancang's dam cascade system. China is already building a new dam (Tuoba: 1039 $Mm^3$) and planning the construction of ten additional ones (MRC, 2020b). All these dams are rather large (e.g., Ru Mei: 13,385 $Mm^3$, Ban Da: 12,902 $Mm^3$, Gu Xue: 10,127 $Mm^3$), and taken together, they have a total storage capacity of about 64,950 $Mm^3$ (Schmitt et al., 2019). If the same filling strategies were to be implemented again, downstream countries should expect a temporary, yet substantial, decrease of water availability, but could also design adaptation and emergency plans. For example, Laos or Cambodia could decide to temporarily change their water management strategies when a new dam becomes operational in the

Lancang. Naturally, information on the past filling strategies could also be used when negotiating the filling of new dams—as for the case of the Grand Ethiopian Renaissance Dam (Zhang et al., 2016; Basheer et al., 2020)—a more desirable and cooperative policy that does not seem to appear at the horizon.

In many ungauged or disputed river basins, like the Mekong, the characterization of hydrological alterations is typically based on 'static' indicators that relate the storage capacity to the average annual discharge volume (Grill et al., 2014, 2015). By coupling actual storage time series with discharge data we can go beyond this first, fundamental, characterization and provide a gateway for a more nuanced understanding of how, and when, reservoir operations affect downstream hydrological processes (Bonnema and Hossain, 2017). In that regard, our results for the Lancang indicate that the fraction of natural flow actively controlled by dams (in northern Thailand and Laos) changes on a monthly basis: reservoirs hold up to ∼50% of the natural flow during the wet season and control almost 89% of the dry season flow coming out of Lancang. Interestingly, we also found that this periodic pattern is not much affected by the hydro-meteorological conditions—like the 2019-2020 drought—partially explaining the complaints and fears of the downstream countries (Eyler and Weatherby, 2020).

From a more technical perspective, another research area that might be influenced by our results is the development of large-scale hydrological models for the Mekong basin. Hydrologists are indeed increasingly interested in the representation of water reservoir storage and operations, a modelling problem that has long relied on generic reservoir release schemes (Hanasaki et al., 2006). Recent research has shown that the nuances of operations at individual dams are better captured by hydrological models when building on high-resolution data available for each dam (Turner et al., 2020). In this regard, we believe our storage and water level time series provide an opportunity for testing and improving the many hydrological models developed for the Mekong basin (Hoang et al., 2019; Yu et al., 2019; Dang et al., 2020a; Yun et al., 2020; Shin et al., 2020; Do et al., 2020). A complementary research direction is the creation of additional datasets for other key variables, such as water temperature or suspended sediment concentrations, which can also be observed, or inferred, from satellite observations (Beveridge et al., 2020; Bonnema et al., 2020; Ahmad et al., 2021).

Like any other numerical modelling study, also this work builds on a few modelling assumptions that are worth being discussed. First, the storage time series we developed have a monthly resolution. As shown in Figure S1, this resolution is sufficient to study reservoir storage dynamics in the Lancang—as well as their impacts on downstream discharge—but it is undoubtedly that the availability of weekly or daily data would further expand the scope of research on transboundary basins. Daily data could be used, for instance, to study the nuances of emergency releases or sudden changes in river discharge. Second, Landsat images provide the best compromise of spatio-temporal resolution and time span, but they do require an image enhancement process. The development of WSA algorithms is an area warranting further research, since studying dams built in the past decades must almost necessarily build on Landsat images. This is a non-negligible factor when considering that many dams have been recently constructed in the tropics, where cloud cover is common during the rain season. Third, the analysis of reservoir filling strategies requires the use of modelled discharge, a problem that could be avoided if riparian countries shared river discharge data. The model validation suggests that the characterization of the filling strategies is not affected by the use of a model, which, importantly, is directly driven by inferred storage data. As mentioned, above the integration of such data with process-based hydrological models is a very active research area that will hopefully lead to even more accurate models.

Looking forward, we should aim at repeating studies like this one at the scale of the entire river basin. Doing that would create a pathway to a robust attribution analysis of the recent droughts that have affected the Mekong countries. It should be 530 noted that such analysis is probably not yet within our reach: we know how runoff generation is spatially distributed (Shin et al., 2020) and we are gathering information on the operations of many reservoirs (Biswas et al., 2021), but we still have limited data on other anthropogenic interventions that arguably affect the overall water balance, such as irrigation activities in the western part of the basin. In turn, this reiterate the need for high-resolution data spanning across countries and socio-economic sectors.

## 6 Conclusions

In just a few decades, the Mekong River basin has undergone a rapid infrastructure development that has fostered economic growth, but also damaged the environment and challenged the relation between riparian countries. A change in this status quo means conceiving cooperative water-energy policies that span across countries and socio-economic sectors (Schmitt et al., 2019; Siala et al., 2021). Aside from the political will, an important piece of the puzzle is the availability of open source datasets 540 that describe how big infrastructures have been operated. Since agreements on data sharing only provide piecemeal information (Johnson, 2020), the use of satellite imagery appears to be the only way to create unbiased observations available to research community and local stakeholders. In this regard, our work complements the existing efforts for the region, bringing us one step closer to a complete understanding of China's management strategies for the Lancang's dams. Importantly, the lessons learnt here could be readily applied to other transboundary river basins, where the lack of information on existing and planned 545 dams is a major obstacle to open science and institutionalized cooperation.

*Author contributions.* D.T.V., T.D.D., and S.G. conceptualized the paper and its scope. Data collection and all analyses were carried out by D.T.V. and S.G.. D.T.V. wrote the manuscript, with substantial inputs from all authors.

*Code and data availability.* The Python scripts used in this study and the corresponding output (E-A-S curves and storage time series) are available at https://github.com/dtvu2205/210520 and https://doi.org/10.5281/zenodo.6299041. The daily discharge data at Chiang Saen 550 were collected from the Mekong River Commission web portal, https://portal.mrcmekong.org/. Observed water level and storage data of Bhumibol and Ubol Ratana reservoirs were collected from the Electricity Generating Authority of Thailand web portal, http://water. egat.co.th/follow.php. CHIRPS-2.0 precipitation data from University of California, Santa Barbara, are available at https://data.chc.ucsb. edu/products/CHIRPS-2.0/. Other input data of the VIC-RES model includes temperature data retrieved from ERA5 dataset (https://cds. climate.copernicus.eu/cdsapp#!/), land use and land cover data obtained from the Global Land Cover Characterization dataset (https://www. 555 usgs.gov/centers/eros/science/), soil data extracted from the Harmonized World Soil database (http://www.fao.org/soils-portal/soil-survey/ soil-maps-and-databases/harmonized-world-soil-database-v12/en/), and the Terra MODIS satellite images (used to calculate the monthly Leaf Area Index and albedo) available at https://earthexplorer.usgs.gov/. The SRTM-DEM is available at https://earthexplorer.usgs.gov/. All

Landsat images used in our study are available at https://earthexplorer.usgs.gov/. The altimetry water level data are retrieved from the Global Reservoirs and Lakes Monitor (G-REALM), https://ipad.fas.usda.gov/cropexplorer/global_reservoir/.

*Competing interests.*  The authors declare that they have no conflict of interest.

*Acknowledgements.*  Dung Trung Vu is supported by the SUTD PhD Fellowship. Thanh Duc Dang and Stefano Galelli are supported by Singapore's Ministry of Education (MoE) through the Tier 2 project 'Linking water availability to hydropower supply—an engineering systems approach' (Award No. MOE2017-T2-1-143). The authors are grateful to Marko Kallio and Maurizio Mazzoleni (and the other two anonymous reviewers) for their insightful comments.

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
