# Peer review of "Satellite observations reveal thirteen years of reservoir filling strategies, operating rules, and hydrological alterations in the Upper Mekong River Basin"

_Hydrology and Earth System Sciences, 2021_

## Author Response (AR1)

Reply to reviewers of paper hess-2021-360

**Satellite observations reveal thirteen years of reservoir filling strategies, operating rules, and hydrological alterations in the Upper Mekong River Basin**

Dung Trung Vu, Thanh Duc Dang, Stefano Galelli, and Faisal Hossain

Resilient Water Systems Group
Pillar of Engineering Systems and Design
SUTD, 8 Somapah Road
Singapore 487372
T. +65 6303 6600
http://people.sutd.edu.sg/~stefano_galelli/

**Editor**

The Referees provided critical comments to the paper. In the discussion phase, the Authors show willingness to address them and improve the description of their work. The revised manuscript should be submitted along with a systematic point-by-point response to all comments.

*We would like to thank the Editor for the positive response and opportunity to revise our work. Following the reviewers' suggestions, we:*

- *Clarified a few aspects related to our methodology (i.e., estimation of the elevation-storage-area curves, analysis of water surface area), as suggested by reviewer #1, #3, and #4;*
- *Moved (and expanded) the description of both hydrological model (VIC-Res) and indicator of hydrological alteration to Section 3.3 and 3.4, as suggested by reviewer #1. In Section 3.3 (and corresponding part of the SI), we also included more information on the validation of VIC-Res;*
- *Provided additional information on the Landsat and altimetry data employed in our study, as recommended by reviewer #3;*
- *Extended the validation of our results by banking on additional water level data that have been recently released on the G-REALM repository (after our manuscript was first submitted). We also added metrics of accuracy, such as CC and RMSE, as suggested by reviewer #3;*
- *Proved that a monthly time step is sufficient for our study. In particular, we provided a comparison between our Landsat-derived water level, altimetry water level (from Jason, which has a 10-day temporal resolution), and Sentinel-1-derived water level (frequency of up to 6 days) for Xiaowan and Nuozhadu reservoirs (which account for almost 90% of the total system storage);*
- *Added a comparison between inferred (from satellite data) and observed storage / water level for two reservoirs (located in the Mekong and Chao Phraya basin) for which such information is available. This validation further highlights the reliability of our methodology. This is a point that was recommended by reviewer #4.*
- *Expanded the discussion on the different causes of the 2019-2020 drought, as suggested by reviewer #2. When doing that, we ensured that our expressions do not lead to any misunderstandings concerning the potential origins of droughts.*

*We would also like to stress that the novelty of our study does not lie in "incremental contributions", as pointed out by reviewer #3. As explained in the Introduction, the novelty of our work stands in three knowledge gaps that we address, that is, (1) lack of water level and storage time series for the Lancang dams, (2) filling strategies of these dams, and (3) event attribution analysis on droughts and pluvials. This is why—we believe—the research is relevant to the special issue on "Socio-hydrology and transboundary rivers". We further stressed this point in our response to reviewer #3. Finally, please note that in our reply-to-reviewers line numbers correspond to the marked-up version of the manuscript.*

**Reply to reviewer #1**

**General Comments:**

Vu et al. write about remotely sensing the filling strategies and operating practices of the Upper Mekong Basin cascade in China. The manuscript highly interesting for actors working in the region. Researchers, NGOs, and state actors in the Lower Mekong Basin should all benefit from understanding the practices of cascade operation in China. The manuscript is very well prepared, and I anticipate it will be highly influential in the Mekong context. I recommend publishing the article subject to some moderate revisions:

*We thank the reviewer for the positive feedback as well as the useful comments for improving the paper.*

**Specific Comments:**

1) Figure 3 presents the workflow in estimating (water level) elevation-storage-area curves. The text tells us that surface area is estimated for every height with one-meter gaps, based on the 30m SRTM DEM. Why, then, is storage estimated with the trapezoidal approximation in Eq. 1? The DEM and elevation bands allows you to directly compute the storage volume, since you already know which pixels fall into which elevation band, and you know the elevation of each pixel. The storage volume is then easy to compute. The trapezoidal approximation is, of course, useful for Manwan Dam.

*This is a good point—thanks for raising it. We decided to use the trapezoidal approximation for the following reasons. First, we would like to minimize the differences in data processing for all reservoirs, including Manwan. Second, the E-A curves estimated from the DEM are well in agreement with the water level observations (from altimetry data) and water surface area (from Landsat images). In other words, these curves are validated. Therefore, we can confidently develop the E-S and A-S curves from the E-A curves using the trapezoidal approximation, which was widely used in the previous studies (e.g., Gao et al., 2012; Bonnema and Hossain, 2019; Li et al., 2019; Tortini et al., 2020). Meanwhile, we do not have observed storage data to validate the E-S and A-S curves estimated directly from the DEM. We included this explanation in the second paragraph of Section 3.1. Please refer to line 200-205 (page 9) of the marked-up manuscript.*

*To corroborate the aforementioned points, we compared the results obtained with the two methods. The differences in storage corresponding to each water level in the variation range are not more than 1% (for Jinghong, Manwan, Miaowei, Huangdeng, and Wunonglong) and 2% (for Nuozhadu, Dachaoshan, Xiaowan, Gongguoqiao, and Dahuaqiao). We show the detailed comparisons for Nuozhadu and Xiaowan reservoirs below. Because the difference is negligible (and our modelling choice is now explained in Section 3.1), we preferred not to include such comparison in the manuscript.*

[Figure]

*Figure 1.1. E-S curve for Nouzhadu and Xiaowan reservoirs obtained via trapezoidal approximation and direct calculation from the DEM.*

|  | Nuozhadu | | | | Xiaowan | | | | | | |
|---|---|---|---|---|---|---|---|---|---|---|---|
| Water Level (m) | Storage [1] (MCM) | Storage [2] (MCM) | Difference | Water Level (m) | Storage [1] (MCM) | Storage [2] (MCM) | Difference | Water Level (m) | Storage [1] (MCM) | Storage [2] (MCM) | Difference |
| 766 | 10501 | 10678 | 1.67% | 1162 | 4077 | 4149 | 1.74% | 1210 | 9112 | 9251 | 1.50% |
| 768 | 10859 | 11042 | 1.65% | 1164 | 4223 | 4298 | 1.74% | 1212 | 9392 | 9534 | 1.49% |
| 770 | 11227 | 11414 | 1.64% | 1166 | 4375 | 4452 | 1.73% | 1214 | 9678 | 9823 | 1.47% |
| 772 | 11605 | 11797 | 1.63% | 1168 | 4531 | 4611 | 1.74% | 1216 | 9970 | 10118 | 1.46% |
| 774 | 11992 | 12189 | 1.62% | 1170 | 4693 | 4776 | 1.74% | 1218 | 10268 | 10419 | 1.45% |
| 776 | 12390 | 12592 | 1.61% | 1172 | 4862 | 4948 | 1.74% | 1220 | 10572 | 10726 | 1.44% |
| 778 | 12798 | 13005 | 1.59% | 1174 | 5036 | 5126 | 1.74% | 1222 | 10882 | 11039 | 1.42% |
| 780 | 13216 | 13428 | 1.58% | 1176 | 5217 | 5309 | 1.74% | 1224 | 11198 | 11358 | 1.41% |
| 782 | 13645 | 13862 | 1.57% | 1178 | 5403 | 5498 | 1.73% | 1226 | 11521 | 11684 | 1.40% |
| 784 | 14084 | 14307 | 1.56% | 1180 | 5595 | 5692 | 1.71% | 1228 | 11849 | 12015 | 1.38% |
| 786 | 14534 | 14763 | 1.55% | 1182 | 5792 | 5892 | 1.70% | 1230 | 12184 | 12353 | 1.37% |
| 788 | 14995 | 15230 | 1.54% | 1184 | 5994 | 6096 | 1.68% | 1232 | 12525 | 12697 | 1.36% |
| 790 | 15468 | 15709 | 1.53% | 1186 | 6201 | 6306 | 1.67% | 1234 | 12872 | 13047 | 1.35% |
| 792 | 15953 | 16199 | 1.52% | 1188 | 6413 | 6520 | 1.65% | 1236 | 13225 | 13404 | 1.33% |
| 794 | 16450 | 16702 | 1.51% | 1190 | 6630 | 6741 | 1.64% | 1238 | 13584 | 13766 | 1.32% |
| 796 | 16958 | 17217 | 1.50% | 1192 | 6853 | 6966 | 1.62% | 1240 | 13950 | 14134 | 1.30% |
| 798 | 17479 | 17743 | 1.49% | 1194 | 7081 | 7197 | 1.61% | 1242 | 14321 | 14508 | 1.29% |
| 800 | 18012 | 18283 | 1.48% | 1196 | 7316 | 7434 | 1.60% |  |  |  |  |
| 802 | 18557 | 18834 | 1.47% | 1198 | 7555 | 7677 | 1.59% |  |  |  |  |
| 804 | 19115 | 19399 | 1.46% | 1200 | 7801 | 7925 | 1.57% |  |  |  |  |
| 806 | 19686 | 19975 | 1.45% | 1202 | 8052 | 8179 | 1.56% |  |  |  |  |
| 808 | 20269 | 20565 | 1.44% | 1204 | 8308 | 8438 | 1.54% |  |  |  |  |
| 810 | 20865 | 21167 | 1.43% | 1206 | 8570 | 8703 | 1.53% |  |  |  |  |
| 812 | 21473 | 21781 | 1.42% | 1208 | 8838 | 8974 | 1.51% |  |  |  |  |

*Storage [1] – obtained by using trapezoidal approximation, Storage [2] – obtained by using direct calculation from the DEM*

2) The methodology regarding determining the surface area from the satellite images seem complex and I'm having trouble following the entire procedure, although the procedure is also presented in Figure 4. Fig. 4 is difficult for me to follow because the analysis has many paths, and for the conclusion I also have to turn around and move upwards in the path. Could you reorganise the figure and leave more space between different paths? E.g., the more space between the path through 1.1 -> 1.5 and 2.1 -> 2.4, so that they are clearly separate? And for the final loop between 2.4 -> 2.6, more space between the boxes. And if it is possible, the outcome could be in the bottom. The general direction in the figure is from top to bottom, but the reversion makes it somewhat difficult to follow. Further about this, I had to read the textual explanation several times before fully comprehending. I'll leave it up to you to decide whether the text needs clarification, as I admit my miscomprehension might be on me alone.

*We agree with you, and we reorganized Figure 4 following these suggestions. The revised version of the figure is reported below.*

[Figure]

*Figure 1.2.WSA estimation algorithm.*

3) Uncertainty quantification? The methodology suggests that there is uncertainty in water pixel identification. In Step 2.5, all pixels within the water cluster are assumed to be water. But substantial amount of pixels in the water cluster in fig 6b should not be assumed to be inundated, particularly in the first two zones with less than 80% of pixels un-inundated. Instead, this reflects a possibility to quantify uncertainty in your methodology.

*We grouped the pixels into 50 zones by their inundation probability based on the frequency map (calculated with the cloudless images). Since the pixels in the same zone have the same*

*(or very similar) inundation probability, there can be two scenarios at each observation time: (1) they are both non-water pixels or (2) they are both water pixels (even when the water fraction of that zone is less than 100%, due to cloud cover). When a zone is identified as a water zone (by k-means clustering in Step 2.5), all pixels in that zone are converted to water pixels. That is the reason for converting all pixels from Zone 14 onwards to water pixels (Figure 6b).*

*There can be a small error in Zone 14--which contains pixels with 26-28% inundation probability--when the threshold of inundation probability is not exactly 26% (e.g., 26.5%, 27%, ...). We could increase the performance by dividing the frequency map into a larger number of zones, but this requires a larger number of cloudless images. More important, our results in Figure 8 show that the WSA estimation algorithm (with 50 zones) works well enough. We clarified these aspects in Step 2.5, Section 3.2. Please refer to line 284-290 (page 12) of the marked-up manuscript.*

What are the underlying elevation values in those zones which fall into water pixels (and non-water pixels)?

*Because of the aforementioned procedure, we do not need to calculate the underlying elevation values in the zones that fall into water pixels (and non-water pixels).*

4) The above leads me to the question, why such a complex procedure? A simpler alternative could be to 1) identify the water pixels as you've done up until step 2.3. 2) With those water pixels you should be able to extract the elevation values at the boundary of the water feature, and 3) this should give you a range of elevation values at the reservoir shoreline. This is the range of possible water elevations (and thus area and volume) which would be easy to communicate in figures too. This method would not require a cloudless image, similarly to the one you're using in the manuscript, but cloudy images would have a smaller number of values at the boundary. I admit that I've not done this and so I don't know what complications there may be, and therefore I do not require that you should do this. But I'd like to see a justification for your choice of method over this simpler alternative.

*We considered the method that you suggested at the beginning of our work. However, the reasons outlined below have prevented us from using it:*

- *First, it is not possible to extract true elevation values from water pixels derived from Landsat images when the water level is below the level corresponding to the SRTM-DEM observation time. In our work, the matter applies to Manwan.*
- *Second, it is difficult to identify the starting point of water surface of the reservoirs in the Lancang. These reservoirs have a long and horizontally narrow shape (see the length in Figure 2 and the shape in Figure S4-S8), so there is not a sudden opening point like in more "regular" reservoirs. Instead, the starting point of water surface moves along the longitudinal direction of the reservoirs. In low flow conditions, the part above the starting point (having higher values of elevation) behaves like a river instead of a portion of the reservoir. In sum, it is not easy to identify the starting point from Landsat images—and this in turn affects the range of elevation values at the reservoir shoreline. This process is even more complicated when dealing with cloud cover and reservoir branches, such as Xiaowan (see Figure S5).*

*Because of these reasons, we developed the idea of identifying the misclassified water pixels (due to cloud cover and other disturbances) based on their inundation probability, which was used before by Gao et al. (2012) and Zhang et al. (2014). The method we used can work well with reservoirs like Manwan. Figure 8 shows that water surface area estimated by our algorithm is well in agreement with the one calculated through the altimetry data.*

5) Your overall methodology is similar to that of the Mekong Dam Monitor. I'd like to see a _short_ comparison of how yours differ from theirs but in more detail than just their choice of using Sentinel and thus having a shorter timeseries (line 54).

*Yes, the overall methodology is similar, in the sense that both our methodology and the one used in the Mekong Dam Monitor (MDM) are based on the idea of extracting the water extent of the reservoirs from satellite images and then converting it into water level and storage by using the information from DEM data. However, there are a few differences:*

- *First, we use the image improvement algorithm, which is important and necessary because it enables us to extract the information on reservoir storage from Landsat images for a long period (2008-now). Meanwhile, to avoid the cloud contamination in satellite images, MDM looks to other remote sensing product, Sentinel-SAR (Synthetic Aperture Radar), which can "pierce" through clouds. However, Sentinels were launched recently (in April 2014), so the information before that time (including the construction and filling periods of five reservoirs on the mainstream of the Lancang) cannot be revealed.*
- *Second, with the water extent estimation provided by our algorithm, we directly infer water level and storage through the E-A-S curves estimated from the DEM. Meanwhile, MDM calculates the average elevation at the reservoir shoreline, then converts it into storage. This way may not work well for all Landsat images (please refer to our previous response).*
- *Finally, to strengthen our results, we make use of water level from Altimetry data (where available) to validate the results obtained by processing the Landsat images.*

*We feel that adding these points to the Introduction (where we talk about the MDM) may slow down the narrative, so we preferred to include the comparison in the SI (Text S1).*

6) Despite my remarks of the methods, the results section is impressive and very useful.

*Thank you.*

7) Section 4.3.2 gives additional theory and methodology with the storage equation, computing evaporation, VIC-Res related methodology etc. I would find it clearer if this methodology would be explained before the results section. The same applies also for indicator of hydrological alteration in section 4.4.

*As suggested, we moved the description of both VIC-Res and indicator of hydrological alteration to Section 3 (before the results section). Please refer to the newly-added Section 3.3 and 3.4. We also took this opportunity to further expand and elaborate on both model and indicator.*

8) I find the indicator of hydrological alteration very clever, as it does not require estimating inflow to the reservoir. However, it does require estimating the streamflow originating from below the cascade. Räsänen et al 2017 estimate the annual inflow to Jinhong to be 58km3 (1840 m3/s), while Chiang Saen annual runoff is 85.5 km3 (from observation timeseries). This is a substantial difference, and needs to be taken into account in computing I. With the VIC-Res already set up, it should not be a big deal. It will be interesting to see how index I changes after accounting for this.

*Our understanding is that the recommendation is to calculate the index not only at Chiang Saen, but also downstream of Jinghong reservoir. We thus proceed to calculate the indicator at this location and reported it below. As one might expect, the analysis shows that the reservoir network exerts a stronger control on the discharge at Jinhong (as compared to Chiang Saen), since the former is located right downstream of the reservoir network. In other words, the difference in impact between Jinhong and Chiang Saen is attributable to the lateral inflow between the two locations, as mentioned by the reviewer. When looking at this analysis, please consider that we made a small modification in the calculation of the indicator (for negative values only), as explained in Section 3.4. Finally, we believe that adding this analysis to the paper may deviate a bit from the main narrative (while further extending its length), so we preferred to leave it out of the revised manuscript.*

[Figure]

*Figure 1.3. The indicator of hydrological calculated downstream of Jinghong reservoir.*

I ask you this because your study deals with a highly political issue, and it is necessary to have good evidence for the statement that China did not change their operating practices despite a severe drought downstream. It would therefore be important to provide validation for the performance of VIC-Res e.g., in the supplementary materials. You point to Dang et al 2020, which gives some validation but does not include the period with Xiaowan and Nuozhadu, and it isn't easy to say how is the performance during the wet season, the time when reservoirs are filled.

*In the previous version of manuscript, the only validation of VIC-Res we provided is a comparison of the simulated and observed storage of Nuozhadu and Xiaowan (Figure S10). We agree that such validation is not very comprehensive (especially when seen in light of the geopolitical implications of our findings), so we proceeded by extending it. In particular, we report below a quantitative comparison of the observed and simulated discharge at Chiang Sean station for the period 2009-2013. This is the only period for which we need VIC-Res (for estimating θ, the fraction of inflow volume retained by the reservoirs during their filling periods). Considering the length of the manuscript, we added this part to the Supplement (Figure S9).*

[Figure]

*Figure 1.4. Comparison between observed and simulated discharge at Chiang Sean station for the period 2009-2013, during which Xiaowan and Nuozhadu were filled in. R, NSE, and TRMSE refer to Correlation Coefficient, Nash-Sutcliffe Efficiency, and Box-Cox Transformed Root Mean Squared Error.*

9) As my last point, I'd like to invite the authors to deposit their results in some open repository (e.g., Zenodo?). The methodology is explained in detail which allows for replication - but since you've already done the work, it would be a great service to the Mekong community to have access to the data - i.e., the water level-storage-area timeseries, maximum and minimum reservoir shapes etc. This would improve the usefulness of the work even further.

*Our results (E-A-S curves and storage time series) and code are already available online, as stated in the "Code and data availability" section (https://github.com/dtvu2205/210520). This said, we reckon it is better to archive the data and get a corresponding doi. We deposited them on Zenodo and added the link (https://doi.org/10.5281/zenodo.6299041) to the "Code and data availability" section (page 25-26).*

**Reply to reviewer #2**

**General Comments:**

This paper aims at assessing the reservoir release rules and downstream discharge of ten large reservoirs along the Lancang River reach by using DEM data, Landsat images, and altimetry data. These data are used to identify elevation-storage and area-storage curves, generate monthly time series of water surface area, and validate the results. I found the study really interesting and well written. Overalls the paper is well structured and with a solid method based on established post-processing approaches for remote sensors data. I think that the paper could be accepted after a moderate revision. Below are my main comments:

*We thank the reviewer for the positive feedback as well as the useful comments for improving the paper.*

**Specific Comments:**

1) One of the main results reported in the abstract is that "two reservoirs were filled in only two years, and that their operations did not change in response to the drought that occurred in the region in 2019-2020". However, this issue is barely discussed in the paper (last paragraph of section 4). Tiezzi (2016) and Hecht et al. (2019) showed that emergency releases from upstream reservoirs could mitigate severe drought in the downstream countries of the Mekong basin in March 2016. Why this is not the case for the drought event that occurred in the period 2019-2020? What is the reason? What is the influence of changes in human presence within the river basin during that drought period on hydropower consumption?

*There could be different reasons behind these divergent management strategies (adopted in 2016 and 2019-2020), such as the inability to reach a political agreement between the riparian countries or the need of following a given hydropower production schedule. Unfortunately, such information is not available in any form, so what we can offer is a plausible explanation based on the timing and magnitude of the two droughts. The 2016 drought had limited magnitude and occurred during the first half of the year, when the reservoir system was releasing water following its normal operations. Right after that, the monsoon season arrived with a relatively high rainfall contribution (please refer to the monthly precipitation anomaly in the Lancang River Basin, Figure 11 (a)). Therefore, the concomitance of monsoon season and emergency releases helped alleviate the drought in the downstream countries. Differently, the 2019 drought had greater magnitude and occurred from the second half of the year, when the reservoir system was storing water. The release of water during the subsequent dry season only partially alleviated the effect of the ongoing drought, since the low precipitation period persisted until mid-2020. We extended our discussion on these findings in the last paragraph of Section 4.4. In this paragraph, we now cite Keovilignavong et al. (2021), who qualitatively reviewed the causes of the Mekong drought before and during 2019–20 and explained the importance of retrieving / sharing information on reservoir operations (which can help characterize the origins of droughts). Please refer to line 471-476 (page 23) of the marked-up manuscript.*

2) The authors used the VIC-Res model developed in Dang et al. (2020) to assess the inflow to the reservoir (Eq.3) to then assess the parameter θ. The first upstream reservoir considered in your study is Wunonglong, which is downstream of the reservoirs Guodo and Jinghe

considered in Dang et al. (2020) (Figure 1). I was wondering how the non-optimal estimation of the streamflow values from the VIC-Res model, based on rule curves conceived to maximize the hydropower production (similarity to Piman et al., 2012), for the Guodo and Jinghe reservoirs may have affected the inflow to the downstream reservoir of Wunonglong. An uncertain estimation of the inflow could lead to an uncertain estimation of the reservoir release (parameter θ). Do you think these may significantly affect the outcome of your study? Is there a way to compare the simulated streamflow with observed values?

*In our study, we calculated the fraction θ of the filling period for the two largest reservoirs, Xiaowan (2009-2010) and Nuozhadu (2012-2013). For that estimation, it is true that the non-optimal estimation of the streamflow values for the Guodo and Jinghe reservoirs may have an effect on the estimated inflow to the downstream reservoirs. However, there are reasons to believe that their effect is marginal. First, Guoduo reservoir joined the system in 2015, after the filling period of both Xiaowan and Nouzhado. Second, Jinghe reservoir—which joined the system in 2004—is located on the Sequ Qu River (a tributary of the Lancang River) instead of the mainstream and, most important, has a capacity of 4 MCM only, while the monthly inflow to Xiaowan varies from about 500 to more than 7500 MCM. Also, note that during the filling period of Xiaowan, all mainstream reservoirs in the upstream of Xiaowan did not exist yet. As for the comparison between simulated and observed streamflow, we note that the VIC-Res model was validated with observed discharge at Jiuzhou station, located right upstream of Xiaowan reservoir.*

*In our initial response to the reviewer (published on line on January 10), we indicated the intention of adding such explanation to the revised manuscript. Having now revised the paper and expanded the description of VIC-Res, we believe that such explanation may not add much to the model description, so we preferred to leave it out.*

3) Have you compared the simulated release from the VIC-Res model (Dang et al., 2020) based on rule curves conceived to maximize the hydropower production (Piman et al., 2012) with the reservoir's releases estimated in your study?

*To answer this question, let us first explain how VIC-Res was used in our study—something we only partially accomplished in our first version of the manuscript. In Dang et al. (2020a), we presented a hydrological-water management model that simulates not only hydrological processes (evapotranspiration, infiltration, baseflow, and runoff) but also the streamflow routing and storage dynamics of each reservoir. As for the dams, we (1) considered the ones built before 2005 and (2) used the rule curves proposed by Piman et al. (2012). That poses two challenges for our current study, since we now: (1) consider more reservoirs (all dams built until 2020) and (2) have the actual storage data retrieved from satellite data. To setup VIC-Res in our study, we therefore proceeded as follows. For each reservoir, we take data on inflow (simulated), storage (estimated from the satellite data), and evaporation (simulated) and invert the mass balance equation to calculate the release. That release time series is used as input to VIC-Res (Dang et al., 2020b) to simulate the storage dynamics of each reservoir. The process is repeated sequentially—starting with the most upstream dam—so as to ensure that the cascading impacts of dams are captured correctly. Because our simulated release is 'driven' by the observed storage, we believe it may not be relevant to compare it against the one obtained in Dang et al. (2020a), where we used the rule curves introduced by Piman et al. (2012). We provided all this information in Section 3.3 of the marked-up manuscript. Please refer to line 328-335(page 14-15).*

4) It is mentioned in section 2 that MODIS data were not considered as "may not be best suited for this study". Indeed, MODIS imagery has high frequency (twice a day) but lower spatial resolution (250 m), which makes it unsuitable for estimating the water surface area of narrow reservoirs, as the case for the Nuozhadu and Xiaowan reservoirs with width between 1000m to 1500m. However, is this the case also for the remaining 8 reservoirs?

*Yes, the remaining eight reservoirs have even smaller surface areas and narrower widths than Nuozhadu and Xiaowan reservoirs. We clarified this point in Section 2.2.2 of the marked-up manuscript. Please refer to line 152-153 (page 7).*

Would it be more beneficial to use MODIS (high frequency but slightly coarser spatial resolution) rather than Landsat images (higher spatial resolution but low temporal frequency) to catch finer fluctuations of reservoirs releases over time?

*This is an option we considered. However, the lower spatial resolution of MODIS makes it unsuitable to capture changes in the water surface area of these reservoirs. In turn, that is likely to lead to higher uncertainties in the water surface estimation process. Because of this reason, we preferred Landsat images (lower temporal resolution, but higher spatial resolution) over MODIS images.*

5) Have you compared the WAS results with the water surface area from Pekel et al. (2016)? They also used Landsat images for assessing global surface water. This comparison would further strengthen your method and the results of your study. You could include this validation in the supplementary material.

*Thanks for your suggestion. We actually considered using the monthly water surface dataset developed by the European Commission's Joint Research Centre (Pekel et al., 2016) to directly infer reservoir operations. However, that dataset is still partially affected by clouds and other disturbances—please refer to the figure below for an example. Naturally, those features do not act as limitations in Pekel et al. (2016), since that study is carried out at the global scale, but are a non-negligible challenge for the goals of our work. In fact, this is why we resort to a specific algorithm for estimating the water surface area. Because of this reason, we believe that adding such comparison may not add much to the validation of our results. For further details, please refer to our response to reviewer #3, comment #1.*

[Figure]

*Figure 2.1. Water extent of Nuozhado (left) and Jinghong (right) reservoirs extracted from Landsat observations in September 2009 by the European Commission's Joint Research Centre (Pekel et al., 2016). Water detection results are affected by clouds and other disturbances such as the no-data stripes in Landsat 7.*

6) Could you summarize the limitations of this study and include them in the discussion?

*As suggested, we summarized and discussed about the limitations of our study in Section 5 (Discussion). We focussed primarily on temporal resolution, image enhancement process, and the reliance on modelled discharge data for the analysis of reservoir filling strategies. Please refer to page 24-25 of the marked-up manuscript.*

**Reply to reviewer #3**

**General Comments:**

Dung Trung Vu and co-authors used Landsat data to derive water area, elevation, and storage series of ten major reservoirs on the main stem of the Lancang River basin. In addition, the authors used a hydrological model to simulate the inflow of the reservoirs and discussed the impact of reservoir filling and operation on the discharge of the Lancang-Mekong River. The authors did provide some suggestions on how to use remote sensing data to obtain reservoir storage changes and combine them with hydrological models to examine the impact of reservoirs. However, the novelty of this study lies in the incremental contributions that are not enough to be considered for publication in the prestigious journal of HESS. In particular, this study falls short of assessing the accuracy and precision of the results. Limitations of this method could impede the application of this approach to other areas. Therefore, I recommend rejection of this manuscript.

*Thank you for taking the time to review our work and provide many thorough comments. We agree that there are opportunities for improving the quality of this work, but we respectfully disagree with the overall evaluation. First, it is true that our methodological approach lies on the advancement of previous works (in particular, Gao et al., 2012 and Zhang et al., 2014), but we do not claim that the novelty of this work lies in its methodological contribution. Instead, the novelty of this study lies in three knowledge gaps that we address, that is, (1) lack of water level and storage time series for the Lancang dams, (2) filling strategies of these dams, and (3) event attribution analysis on droughts and pluvials (please refer to the last two paragraphs of the Introduction). This is why—we believe—the research is relevant to the special issue on "Socio-hydrology and transboundary rivers". Second, the accuracy and precision of the results can be further evaluated by banking on a few additional datasets, as explained below. Finally, we disagree that our methodological approach cannot be applied to other areas, since (1) all datasets have global coverage, (2) both data and code are publicly available, and (3) we contributed an algorithm for improving the water surface estimation of Landsat images—something that makes them more usable, especially in regions affected by cloud cover.*

**Specific Comments:**

1) The water surface area extraction algorithm needs substantial improvement. There are already many decent global water surface area datasets with spatial and temporal resolutions that fully meet the requirements of the study and their algorithms are relatively advanced (Pekel et al. 2016; Pickens et al. 2020). These data can at least be used as input data to generate Water Layers instead of simply using NDWI with a fixed threshold.

*The key benefit of using the Pekel et al. (2016) or Pickens et al. (2020) method is that these algorithms provide a global product on water surface area with a probability of occurrence of water for every grid cell based on long-term Landsat record. Based on this long-term record of Landsat data during cloudless days (and without the challenging issues of speckle and shadows), these products essentially inform the user of the probability of a given grid cell being water at a given time of the year, which is particularly useful if there are clouds present---thus the issue of cloud cover can be mitigated. There have been many studies that have improved water area classification for reservoirs by building on such studies by Pekel*

*et al. (2016) or Pickens et al. (2020). For example, Zhao and Gao (2018), uses Pekel's dataset (which has recently been updated to 2020), to improve water classification during cloudy or challenging situations common in Southeast Asia. Regardless of what specific water area classification method is used (e.g., index based such as NDWI, MNDWI), the key source of uncertainty in water area estimation using visible sensors (Landsat) will always be that of cloud cover and, when the various methods are compared, we have found NDWI to perform more robustly in Asian climates. Please refer to our previous studies (e.g., Biswas et al. 2021a, b).*

*In our study, we are following exactly a similar procedure, one that builds not only on an index classification but also on the prior probability of water occurrence (please refer to lines 243-254 and Figure 5). Thus, our method is already inclusive of the Pekel et al.'s dataset concept and is not a pure index-based classification—this ensures that our water surface area estimates are robust during cloud cover situations. Moreover, the reviewer should note that, in our experience, Pekel et al.'s dataset suffers from a few problems, despite the use of a long-term Landsat record, even during cloudy days and for smaller water bodies and in steep terrain (please refer to the figure below). For example, we have frequently found Pekel's data-based water classification (such as Zhao and Gao, 2018) to perform poorly in patches for reservoirs in Mekong, requiring therefore additional sensors and more creative methods, which our study has incorporated. Finally, our study employs k-means clustering as the final filter to further improve water area estimation (see Figure 6). In sum, our water surface area is a robust approach involving three layers of improvement—prior probability, index classification, and k-means clustering.*

[Figure]

*Figure 3.1. Water extent of Nuozhado (left) and Jinghong (right) reservoirs extracted from Landsat observations in September 2009 by the European Commission's Joint Research Centre (Pekel et al., 2016). Water detection results are affected by clouds and other disturbances such as the no-data stripes in Landsat 7.*

[Figure]

*Figure 3.2. Water extent of Nuozhado (left) and Jinghong (right) reservoirs extracted from Landsat observations in September 2009 (downloaded from the dataset of Pickens et al. (2020)). Water detection results are heavily affected by clouds and other disturbances such as the no-data stripes in Landsat 7.*

2) The author's introduction of the input data is not clear enough. Different levels of input Landsat data (TOA reflectance/surface reflectance/DN) may yield different water body index results, so the level of input data needs to be explicitly shown.

*For the Landsat data, we used the Landsat Collection 1 Level-2 (Surface Reflectance) downloaded from the USGS website (https://earthexplorer.usgs.gov/). This is because we are doing a retrospective analysis and not developing an operational tool that needs near real-time data. (In fact, no study should really use TOA reflectance unless it is for some kind of real-time data assimilation or if fast updates are needed.) We clarified this point in Section 2.2.2 of the marked-up manuscript. Please refer to line 148-149 (page 7).*

Also, the type of Jason data needs to be shown and the processing of extracting the water body elevations from the altimetry data needs to be introduced since whether and which waveform retracking algorithm used should largely affect the results. Without really showing the waveform retracking algorithm and specific thresholds for the classic pulse limited radar altimeters, these results are highly unconvincing.

*For the radar altimetry, we used the data provided in the G-REALM repository (https://ipad.fas.usda.gov/cropexplorer/global_reservoir/), which is based on the analysis of Jason-2,3 and Sentinel-3A,B altimetry data. This is a well-known dataset, described in Birkett et al. (2010a, 2010b). We agree that this information should have been provided but that does not mean that the results are "highly unconvincing" since our analysis includes both Landsat-derived and altimetry-converted water surface area. We provided this information in Section 2.2.3 of the marked-up manuscript. Please refer to line 159-161 (page 7).*

3) More data are needed for assessing the water area/elevation/storage results. The authors used Jason-2 and Jason-3 altimetry data to evaluate the water elevations derived from Landsat data in the Xiaowan and Nuozadu reservoirs. However, the time span and sampling of the altimetry data is relatively low at the two reservoirs. For example, altimetry-based water levels for the Nuozadu Reservoir from 2017 to 2020 are missing. Therefore, more validation data should be supplemented, such as water surface elevations derived from

Sentinel-3 and ICESat/ICESat-2 data. In addition, the accuracy and precision of satellite altimetry data need to be supported by validation results.

*Thanks for your suggestion. G-REALM has recently released a few new datasets, so we banked on them to extend the validation. In particular, we extended the time span of water level (obtained from Sentinel 3A) for Nuozhadu (note that water level data are now available for all years, with the exception of 2019 and part of 2020) and added a validation for Huangdeng (2009–2020), and Jinghong (2019-2020 only)–the latter obtained from Sentinel 3B. We thus updated Figure 8, S11, and S12. We also provided a new figure showing the validation for Huangdeng and Jinghong reservoirs. Please find it below or in Supplement (Figure S13). As for the accuracy and precision of the satellite altimetry data, we note that extensive validations were carried out in Birkett et al. (2010a, 2010b) with gauged data. Finally, thank you for suggesting the use of IceSat-2 for further validating our results. However, IceSat-2 has a 91 day repeat and is therefore not useful for understanding or improving reservoir dynamics or operations for the study region.*

[Figure]

*Figure 3.3. Water surface area of Huangdeng (top) and Jinghong (bottom) reservoirs.*

4) The accuracy of the water surface area/elevation/storage results needs to be described in detail. There are no statistical metrics in the manuscript to characterize the accuracy and uncertainty of the results. For example, CC and RMSE can be used to describe the consistency between Landsat-derived elevations and altimetry-derived elevations.

*Thank you for raising this point. We added a table of quantitative comparison of Landsat-derived and altimetry-converted water surface area. Please find it below or in Supplement (Table S6).*

*Table 3.1. Quantitative comparison of Landsat-derived and altimetry-converted water surface area. RMSE is measured in km².*

| Reservoir | R (CC) | RMSE (km²) | NRMSE |
|-----------|--------|------------|-------|
| *Nuozhadu* | *0.994* | *13.941* | *0.049* |
| *Xiaowan* | *0.977* | *9.901* | *0.062* |
| *Huangdeng* | *0.977* | *1.884* | *0.077* |
| *Jinghong* | *0.558* | *0.428* | *0.020* |

5) The temporal resolution of the study results needs to be substantially improved. As can be seen from the presented graphs, the water elevations of the reservoirs change rapidly from June to October, and the monthly water elevation and storage series may not accurately depict the real operation of the reservoirs.

*The temporal resolution is sufficient for studying the reservoir dynamics of many Asian reservoirs subject to the Monsoon. That is because many reservoirs will typically begin filling sometime during the monsoon season (June-July) and drain from November onwards as part of flood control and irrigation requirements (with hydropower needs controlling the rate at times). This is a point demonstrated in Biswas et al. (2021a), who used an entirety of 35 years of Landsat data to show that we can understand reservoir storage dynamics for 1,598 reservoirs with confidence to track the gradual increase/decrease of active storage as well as inter-annual variability.*

*To prove that a monthly time step is sufficient for our study, we provide a comparison between our Landsat-derived water level, altimetry water level (from Jason, which has a 10-day temporal resolution), and Sentinel-1-derived water level for Xiaowan and Nuozhadu reservoirs. The data of Sentinel-1-derived water level have a frequency of up to 6 days (Sentinel-1A and B have a frequency of 12 days and interleave to each other) and were archived from Mekong Dam Monitor Platform. Note that the Sentinel-1-derived water level is available since 2015, but the Mekong Dam Monitor has published data for Xiaowan from 2015 and for Nuozhadu from 2016. However, the data for the very first period are sparse, so we plotted data from 2017 onwards. Overall, the comparison (shown below and in Figure S1 in the Supplement) shows that the use of a monthly resolution yields the same trajectories of a weekly one, so our analysis does depict the real operation of the reservoirs.*

[Figure]

*Figure 3.4. Comparison between Landsat-derived water level (green line), Jason altimetry water level (blue dots), and Sentinel-1-derived water level (orange dashed line) for Nuozhadu (left) and Xiaowan (right) reservoirs.*

6) The advancement of the study results is not shown compared to the hydrological model results. It can be seen from Figure S5 that the reservoir water storages from the satellite

observations and the VIC-Res model simulations are very similar. Therefore, what is the necessity for performing this study that used remote sensing data?

*It is indeed good news that a macro-scale hydrological model can accurately reproduce hydrological and water management processes in the Lancang. However, it is necessary to consider two key elements. First, for basins like the Lancang (for which streamflow and water management data are not publicly available), we need data to setup and validate hydrological models—hence the need for this research, as explained in line 514-523. Second, the accuracy of models simulating water reservoir dynamics can be improved by banking on data providing information on storage dynamics, filling strategies, and rule curves. In our case, for example, the storage data (inferred from satellite data) are used by VIC-Res to solve the mass balance of each reservoir—this is a point we expanded on, as explained to reviewer #2, comment #3. In sum, we believe that the information provided by hydrological models and satellite data is complementary, not interchangeable.*

7) The authors used model simulated inflow (Q) and evaporation loss (E) to calculate the fraction of inflow volume retained by the reservoir (θ). Unfortunately, the authors did not give the method and input data used to obtain E and the share of E in

*In our study, evaporation losses from the reservoir surface are modelled with the Penman equation for all cells belonging to the impoundment—a functionality available in the VIC-Res software (Dang et al., 2020). In the Penman equation, solar radiation and surface wind data were derived from the Global Meterological Forcing Dataset (Sheffield et al., 2006). We clarified this point in Section 3.3 of the manuscript. Finally, we note that the evaporation values for humid basin reservoirs of South and Southeast Asia rarely makes a difference in improving outflow estimation. This has been shown in Bonnema et al. (2017).*

8) For the 2019-2020 drought event, the authors argue that the Lancang dams did not change their operating patterns and stored about 46% of the estimated natural flow during the wet season, and the operations "contributed to downstream droughts and pluvials". However, from another point of view, the reservoirs could retain a fixed percentage of water when inflow decreases, which reduces the storage increment and increases the outflow compared with retaining water to a certain elevation (a commonly used operating rule). Therefore, it cannot simply be considered that "dam operations contributed to downstream droughts and pluvials".

*In our analysis, we simply followed the data, which indicate that part of the river discharge has been withheld during the drought. Naturally, this is not the only cause for the downstream drought, so this is why we use the verb "partially contributed" instead of "caused". Following the suggestion from reviewer #2, we also expanded the discussion on this drought, added a few more key references, and overall explained that the 2019-2020 drought was likely caused by the concomitance of various factors, including precipitation anomalies and reservoir operating strategies.*

9) The authors overstate the impact of new reservoirs on downstream water discharge. According to the authors' calculation, Nuozhadu (21749 Mm3) and Xiaowan (14645 Mm3) reached steady-state operations in about two years by retaining from 15% to 23% of the annual inflow volume. And the newly building reservoir, Tuoba (1039 Mm3), has less than one-tenth of the capacity of Xiaowan or Nuozhadu. But the authors claimed that downstream

countries should expect a temporary, yet substantial, decrease of water availability if the same filling strategies were to be implemented. This does not seem to have a solid basis.

*We disagree with this comment, simply because it is not based on what we wrote. In the second paragraph of Section 5, we wrote that "China is already building a new dam (Tuoba; 1039 Mm3) and planning the construction of ten additional ones (MRC, 2020)", so it is the filling of multiple additional dams that could cause "a temporary, yet substantial, decrease of water availability". Please note that all these dams are rather large: e.g., Ru Mei (13,385 Mm3), Ban Da (12,902 Mm3), Gu Xue (10,127 Mm3); taken together, they have a total storage capacity of about 64,950 Mm3 (Schmitt et al. 2019). We made this point crystal clear in the marked-up manuscript. Please refer to line 494-499 (page 23-24).*

10) The Authors need to pay attention to the citations. For example, the authors did not cite related literature in their initial references to NDVI, NDWI, and MNDWI. Also, the authors did not show specific sources of remote sensing (Landsat/Jason) or other data (CHIRPS-2.0) they used and cite them properly in the text or the supplement content.

*In the first sentence of Section 3.2, we introduce the spectral indices (i.e., NDVI, NDWI, and MNDW) and refer to Table S4, which contains the name, formula, and references for each of them. The specific sources of remote sensing (STRM-DEM/Landsat/Jason) or other data (CHIRPS-2.0) with the links to access were stated in the code and data availability section (page 25-26).*

**Reply to reviewer #4**

**General Comments:**

The manuscript titled "Satellite observations reveal thirteen years of reservoir filling strategies, operating rules, and hydrological alterations in the Upper Mekong River Basin" by Vu et al., simulated the cascade reservoir operation in Upper Mekong River using satellite observations. This study applied the SRTM-DEM, Landsat and Jason Altimetry observations over the study area and inferred the storages variations of two largest reservoirs to assess the reservoir operations against meteorological changes. The manuscript is generally well structured, however, there are concerns regarding the validation of the models/results, which would hinder the reliability of the conclusions made. My main comments are as follows:

*Thank you for these detailed comments, which we will help us strengthen our study. In our response below, we clarify various aspects related to the methodological approach, elucidate on the reliability of methods and results, and explain how the results validation can be further extended.*

**Specific Comments:**

1) SRTM-DEM was used in calculating the reservoir water storage. As stated in the manuscript, the reservoir constructions happened after the year of 2000 whereas SRTM bathymetry measurements was conducted in 2000. Such SRTM-DEM measurements may miss out the potential bathymetry changes caused by local reservoir constructions.

*Thank you for raising this point. Yes, it is true that reservoir construction may change the bathymetry but is also true that these changes are negligible (at least for our study site). That's because of two reasons. First, Lancang's reservoirs have horizontally narrow and long shapes. Their length varies from about 25 km (Dahuaqiao) to about 198 km (see Figure 2). Because of these characteristics, dam construction sites (often carried out near the dam location) only affect a very small portion of the reservoir bathymetry. Second, Lancang's reservoirs have a large portion of dead storage, from about 32% (Xiaowan) to 87% (Wunonglong) (see Table S1 for the specific volumes of the reservoirs). Therefore, we can say that the reservoir bathymetry in the variation range of the reservoirs is barely affected from dam constructions. We stressed this point in the first paragraph of Section 2.2.1. Please refer to line 130-136 (page 5-6) of the marked-up manuscript.*

*To make our results more reliable, we compared the E-A curves estimated from DEM data with the ones obtained by paring altimetry water level and Landsat-derived water surface area—which are not affected by bathymetry changes caused by reservoir constructions. This comparison showed good agreements in the cases of Xiaowan and Nuozhadu (see Figure 7). With more altimetry water level data recently published on G-REALM, we added the comparisons for two other reservoirs Huangdeng and Jinghong (see Figure S11 and S12).*

The spatial resolution of SRTM-DEM is 30m, however, the authors calculate "the surface area corresponding to each 1-m elevation of the DEM" (Page 7), Please explain in more details for the processing procedure.

*SRTM-DEM has a spatial resolution of 30 m that is the actual size of each pixel on the ground, with the value of each pixel being the elevation of the area represented by that pixel. The processing procedure works as follows:*

- *First, we isolate the DEM data with the contour corresponding to maximum water level and dam crest line. The purpose of this step is to calculate the curve within the extent of the reservoir only and thus avoid errors due to the surrounding areas.*
- *Then, we calculate the surface area corresponding to each 1-m elevation of the DEM. Specifically, with each elevation value (each meter) from the lowest elevation within the reservoir extent to the maximum water level, we count the number of pixels having a value equal to or smaller than that elevation value. This is because, when water reaches that elevation, the area corresponding to those pixels is inundated. Then, we multiply the number of pixels by the pixel size (30 m x 30 m) to get the water surface area (on the ground).*
- *Finally, we fit a five-degree polynomial (degree determined by trial-and-error) to the data points so obtained.*

*We included these additional details in the second paragraph of Section 3.1. Please refer to line 183-191 (page 8-9) of the marked-up manuscript.*

2) Landsat dataset is another key to solve water surface area in the article. The biggest challenge for the image interpretation is to distinguish water-covered cells from the non-water areas impacted by cloud and other contributors. Water regions suffered from, or chlorophyll concentration or aquatic plants are not inclined to adopt NDWI as the water index is sensitive to vegetation. Matching to the maximum water extent from Pekel et al., (2016) may be caused by the aqua-vegetated problem. Meanwhile, for the water regions with narrow width, some other researchers are inclined to use MNDWI (Li et al. 2019). This deserves the authors a careful investigation for the local reservoir conditions.

*Thank you for making this point. We are aware that the spectral indices for water surface extraction perform differently in different regions. This is why we carried out our initial assessment on the performance of three commonly used spectral indices (NDVI, NDWI, MNDWI), an assessment that has been further extended (please refer to our response to comment #5). In addition to our comparison with the maximum water extent from Pekel et al., (2016), we manually checked the obtained water layers with the true colour Landsat images before making our decision of using NDWI. We stressed this point in Step 1.2, Section 3.2. Please refer to line 234-243 (page 11) of the marked-up manuscript. Finally, we note that the reviewer did not provide a reference for "Li et al. 2019", so we were not able to rely on this paper for this specific response.*

3) For the water area extraction with cloudlessness, although the pixels are free from cloud, they may still be affected by ground conditions, such as vegetation, deep or shallow bottom, or water turbidity. Setting the water index threshold as a constant 'zero' value may not be reasonable enough to deal with the aforementioned problem. Additionally, the operations of [1.4] and [1.5] tend to artificially increase the water coverage and would cause the total water storage larger than it potential might be. Such operations are lack of a solid theory to support. It might be a little bold to be directly applied over an ungauged basin without observations taken as validation. I would expect the authors could provide more reasons for doing so.

*The water layers obtained from the NDWI-based classification (with a threshold of 0) in Step 1.2 are not used to infer the water surface/storage directly. Instead, they are used to create the zone mask and the expanded mask for Phase 2, where we improve the water classification with a robust NDWI threshold obtained via k-means clustering. Note that the expanded mask (Step 1.4) is used for the reservoir extent isolation, not for inferring the water surface/storage, so it does not artificially increase the water coverage/storage. Also, the expanded mask (Step 1.5) is used for clustering the pixels, not for inferring the water surface/storage. Finally, we would like to stress that our approach is based on a solid theory. In particular, we extend and improve the WSA estimation algorithm introduced by Zhang et al. (2014), which is validated for a few reservoirs in South Asia.*

*Naturally, in our case it is not possible to collect measured water level/storage of Lancang reservoirs. Therefore, we validated our results with reservoir water level from altimetry collected from G-REALM (Birkett et al. 2010a and 2010b). As explained in our response to comment #7, such validation was further extended for Huangdeng and Jinghong reservoir (see Figure S13). Moreover, we validated our methodology on two reservoirs in the Lower Mekong and Chao Phraya Basin, for which storage/water level observations are available to the public. Please find it below or in Supplement (Figure S3).*

[Figure]

*Figure 4.1. Water surface area (a,b) with their statistical metric and storage variations (c,d) of Bhumibol reservoir (left) and Ubol Ratana reservoir.*

4) For the water area extraction with cloud and other disturbances, this article "resorts to k-means clustering". This is interesting approach but its reliability in ungauged area is unsure. Since there lacks a solid theory to support and needs manually adjustments. I would

recommend the authors try OSTU index (a method dynamically obtain a threshold) to compare the difference they may result in threshold calculation as well as in water storage. By doing this, an uncertainty estimation can be given to the reservoir water storage.

*Thank you for your recommendation. As explained above, our methodological approach relies on solid theory and our results have been validated. We considered the OSTU index, but we preferred to exclude it from the analysis—extending the validation of our results was, in our opinion, a better approach towards strengthening our study.*

5) In Section 3.2, the author sates that NDWI is better than MNDWI to infer water surface area of reservoir. However, stating that based on the Xiaowan Reservoir only is insufficient. Could the author explain the reason why the maximum water extent was validated on two reservoirs instead of the ten?

*The reason for which we included the comparison between NDVI, NDWI, and MNDWI for only one reservoir is that our results (based on NDWI) are then validated. This said, we agree with the reviewer that such comparison should be carried out for the other reservoirs. We provided the validation for other reservoirs in Figure S4-S8 in Supplement.*

6) Regarding the WSA estimation algorithm in Figure 4, why were cloudy images taken into account in NDWI calculation to obtain the NDWI Layer? Please specify and explain this.

*During the monsoon season, Landsat observations are heavily affected by clouds. If we use cloudless images only, the (estimated) water surface area (WSA) data pertaining to the monsoon season may therefore be inaccurate/missing. Therefore, the main purpose of developing and using the WSA algorithm is to improve water classification for all images (especially for cloudy images). We stressed this point in Step 2.1, Section 3.2. Please refer to line 258-261(page 11) of the marked-up manuscript.*

7) Validation of the results (Section 4) is too weak. The author only validated water level from the Radar Altimetry data and only two reservoirs have Radar Altimetry data. Furthermore, there is no validation of reservoir storage. This makes the results inconvincible.

*We agree with reviewer that the validation could be strengthened, but that does not mean that the results are "inconvincible". That's because of four reasons. First, the storage of the two validated reservoirs accounts for about 86.45% of the whole system capacity. As illustrated in our analysis, understanding the storage and release dynamics of these two reservoirs is a key step towards explaining the dynamics of the entire system. Second, the reservoir curves for Nuozhadu and Xiaowan are also validated (Figure 7) with altimetry data, so that yields an explicit validation of surface area and an implicit validation of storage. Third, the methodological approach we build on has been adopted for several other sites in Asia and Southeast Asia (e.g., Gao et al., 2012). Fourth, we now include a validation of inferred storage / water level for reservoirs for which this information is available (please refer to our response to comment #3).*

*With more altimetry water level data published on G-REALM recently, we provided in Figure S13 in Supplement, the validation with altimetry water level for a few additional reservoirs, including the third largest one in the system—Huangdeng (3.37% of the whole system capacity). For additional details, please refer to our response to reviewer #3 comment #3.*

8) The author tried to estimate monthly reservoir storage of the ten reservoirs. However, from the results part, we only see the results of the Nuozhado and Xiaowan (Figure 9). Results of other reservoirs are shown in 8 reservoirs. Why didn't the author illustrate monthly reservoir storage of other 8 reservoirs? Please specify and explain.

*The monthly reservoir storage of each reservoir is illustrated in Figure S14. As for Figure 9, we prefer to keep it as is (with the storage of the eight remaining reservoirs aggregated into one time series), because the individual capacity of the eight remaining reservoirs is too small compared to the two largest ones.*

9) In section 4.3.2, the author used VIC-Res, a hydrological model to simulate the inflow of the reservoir. Could the authors explain in more details on the details of the simulation to justify the performance of the model, i.e., input of the model, parameters, calibration, and validation of the results.

*We also received a few questions about VIC-Res from the other reviewers. So, we provided more information about VIC-Res in Section 3.3. Please refer to page 14-15 of the marked-up manuscript.*

[revised manuscript text omitted]